# Global Ensemble of Temperatures over 1850-2018: Quantification of Uncertainties in Observations, Coverage, and Spatial modelling (GETQUOCS)

Maryam Ilyas[a,b], Douglas Nychka[c], Chris Brierley[d], and Serge Guillas[a]

[a]Department of Statistical Science, University College London, London, UK
[b]College of Statistical and Actuarial Sciences, University of the Punjab, Lahore, Pakistan
[c]Applied Mathematics and Statistics, Colorado School of Mines, Golden, CO, USA
[d]Department of Geography, University College London, London, UK

**Correspondence:** maryam.stat@pu.edu.pk, maryam.ilyas.14@ucl.ac.uk

**Abstract.** Instrumental global temperature records are derived from the network of in situ measurements of land and sea surface temperatures. This observational evidence is seen as fundamental to climate science. Therefore, the accuracy of these measurements is of prime importance for the analysis of temperature variability. There are spatial gaps in the distribution of instrumental temperature measurements across the globe. This lack of spatial coverage introduces coverage error. An approximate Bayesian computation based multi-resolution lattice kriging is developed and used to quantify the coverage errors through the variance of the spatial process at multiple spatial scales. It critically accounts for the uncertainties in the parameters of this advanced spatial statistics model itself, thereby providing for the first time a full description of both the spatial coverage uncertainties along with the uncertainties in the modelling of these spatial gaps. These coverage errors are combined with the existing estimates of uncertainties due to observational issues at each station location. It results in an ensemble of 100,000 monthly temperatures fields over the entire globe that samples the combination of coverage, parametric and observational uncertainties from 1850 till 2018 over a $5° \times 5°$ grid.

## 1 Introduction

Instrumental surface temperature data sets are frequently used to determine natural variability of changing surface temperatures on Earth (e.g. Hansen et al., 2010; Morice et al., 2012; Menne et al., 2018). Climate models also use instrumental observations for accurate assessment of various climate phenomena (e.g. Milton and Earnshaw, 2007; Edwards et al., 2011; Glanemann et al., 2020). Temperature data bases are generally created by blending the land and sea surface temperature records. The land component of the data sets is mostly collected from the global historical network of meteorological stations (e.g. Jones et al., 2012). These are mostly derived from the World Meteorological Organization (WMO) and Global Climate Observation System (GCOS) initiatives. On the other hand, sea surface temperatures are largely compiled by the International Comprehensive Ocean-Atmosphere Data Set (ICOADS) (Woodruff et al., 2011). These are collected from ships and drifting buoys (e.g.

Kennedy et al., 2011b).

These raw temperature estimates are post-processed by removing biases from them (Dunn et al., 2014). In a first step of quality control, noise originating from instrumental or observer error is removed (Dunn et al., 2016). After this, systematic biases that arise from station movements or incorrect station merges, changes in instruments and observing practices and land use changes around stations (more commonly known as urbanization impacts) are removed (Dunn et al., 2016). Such a homogenization process (Domonkos and Coll, 2017) aims to remove or at least reduce the non-climatic signals that will likely affect the genuine data characteristics (e.g. Hausfather et al., 2016; Cao et al., 2017).

Blended land and sea surface temperature data are generated by a variety of organizations. These include: Global Surface Temperatures (NOAAGlobalTemp) by the National Oceanic and Atmospheric Administration (NOAA) (Smith et al., 2008; Vose et al., 2012; Zhang et al., 2019; Vose et al., 2021), Goddard Institute for Space Studies (GISS) surface temperature anomalies by the National Aeronautics and Space Administration (NASA) (Hansen et al., 2010; Lenssen et al., 2019), temperature anomalies provided by Japanese Meteorological Agency (JMA) (Ishihara, 2006), HadCRUT5 temperature anomalies by the Met Office Hadley Centre and the University of East Anglia Climatic Research Unit (Morice et al., 2012, 2021), and Berkeley Earth Surface Temperature (BEST) by Rohde and Hausfather (2020). Each group compiles these monthly temperature products using somewhat different input data, and extensively different quality control and homogenization procedures (e.g. Rohde, 2013; Jones, 2016).

The GISS temperatures make substantial use of satellite nightlight data (Hansen et al., 2010; Lenssen et al., 2019) for bias adjustment of urban areas. However, NOAAGlobalTemp, HadCRUT5 and BEST use no satellite data at all (Rohde and Hausfather, 2020; Vose et al., 2021; Morice et al., 2021). These data sets are also different in terms of their starting years: 1850-present for HadCRUT and BEST; 1880-present for GISS and NOAA; and 1891-present for JMA. The spatial resolution is different as well. Each group also employs different methods of averaging to derive gridded temperature products from in situ measurements (Jones, 2016; McKinnon et al., 2017).

In addition to these methodological differences, spatial coverage is also being treated differently by these groups (Huang et al., 2020). The HadCRUT4 dataset do not interpolate over grid boxes having missing observations. The sea component of JMA grid estimates are based on optimally interpolated (i.e. kriging) sea surface temperature anomalies (Ishii et al., 2005; Kennedy, 2014). On the other hand, no spatial interpolation is performed on HadSST3 (Rayner et al., 2006) and CRUTEM4 (Jones et al., 2012) that are the land and sea components of HadCRUT4 data set. The NOAAGlobalTemp (Vose et al., 2021) data set is based on a nonparametric smoothing process and empirical orthogonal teleconnections. The reconstruction combines low frequency spatial running average and a high frequency reduced space analysis. For broader spatial coverage, the GISS uses a linear distance weighting with data from all the stations up to 1200 km of the prediction location (Hansen et al., 2010). The weight of each sample point decreases linearly from unity to zero. This interpolation scheme computes estimates

by weighting the sample points closer to the prediction location greater than those farther away without considering the degree of autocorrelation for those distances. On the other hand, the JMA (Ishii et al., 2005) ocean records use covariance structure of spatial data and are based on traditional kriging. Formal Gaussian process regression is used by the BEST to produce spatially complete temperature estimates (Rohde et al., 2013). Cowtan and Way (2014) also handle the the issue of missing observations and provide a data product that is based on HadCRUT4 temperature estimates (Morice et al., 2012). This data set (Cowtan and Way, 2014) consists of spatially dense fields. The unobserved grid cells of HadCRUT4 spatial fields are estimated using a spatial interpolation approach i.e. ordinary kriging.

Recently, a new monthly temperature data set was created (Ilyas et al., 2017). It employs the multi-resolution lattice kriging approach (Nychka et al., 2015) that captures variation at multiple scales of the spatial process. This multi-resolution model quantifies gridded uncertainties in global temperatures due to the gaps in spatial coverage. It results in a 10,000 member ensemble of monthly temperatures over the entire globe. These are spatially dense equally plausible fields that sample the combination of observational and coverage uncertainties. The data are for open access and freely available at: https://oasishub.co/dataset/global-monthly-temperature-ensemble-1850-to-2016.

This paper provides a substantial update on Ilyas et al. (2017) data set. Here, a new version of this data set is produced that incorporates the uncertainties in the statistical modelling itself (i.e. parametric uncertainties) in addition to the observational and coverage errors. To account for the model parametric uncertainties, an approximate Bayesian inference methodology is proposed that extends the multi-resolution lattice kriging (Nychka et al., 2015). It is based on the variogram, a measure of spatial variability between spatial observations as a function of spatial distance. However, it is important to note that all the components of the HadCRUT4 uncertainty model are not used in this study and only the uncertainties encoded in the HadCRUT4 ensemble members are used.

## 2   Methods

### 2.1   Multi-resolution lattice kriging using ABC

The Multi-resolution lattice kriging (MRLK) model was introduced by Nychka et al. (2015). It models spatial observations as a sum of a Gaussian process, a linear trend and a measurement error term. The MRLK can flexibly adjust to complicated shapes of the spatial domain and has the property of approximating standard covariance functions. This methodology extends spatial methods to very large data sets accounting for all the scales, for the goals of spatial inference and prediction. Indeed, it is computationally efficient for large data sets by exploiting sparsity in precision matrices. The underlying spatial process is a sum of independent processes, each of which is a linear combination of the chosen basis functions. The basis functions are fixed and coefficients of the basis functions are random.

Consider observations $y(\mathbf{x})$ at $n$ spatial locations $\mathbf{x}_1, \mathbf{x}_2, ..., \mathbf{x}_n$ in the spatial domain $D$. The aim is to predict the underlying process at an arbitrary location $\mathbf{x} \in D$ and to estimate the uncertainty in the prediction. For $\mathbf{x} \in D$,

$$y(\mathbf{x}) = d + g(\mathbf{x}) + \epsilon(\mathbf{x}) \tag{1}$$

where $d$ is the mean and $\epsilon$ is the error term. The unknown spatial process $g(\mathbf{x})$ is assumed to be the sum of $L$ independent processes having different scales of spatial dependence. Each process is a linear combination of $m$ basis functions where $m(l)$ is the number of basis functions at level $l$,

$$g(\mathbf{x}) = \sum_{l=1}^{L} g_l(\mathbf{x}) = \sum_{l=1}^{L} \sum_{j=1}^{m(l)} c_j^l \phi_{j,l}(\mathbf{x}) \tag{2}$$

The basis functions ($\phi_{j,l}$) are fixed. These are constructed at each level using the unimodal and symmetric radial basis functions. Radial basis functions are functions that depend only on the distance from the center.

The inference methodology (Nychka et al., 2015) is the direct consequence of maximizing the likelihood function. This inference framework does not account for the uncertainty in the model parameters within the MRLK (Nychka et al., 2015). Here, we estimate the MRLK parameters and quantify uncertainty in these parameters. For this purpose, a Bayesian framework is created in which the posterior densities of the multi-resolution lattice kriging parameters are estimated using the Approximate Bayesian Computation (ABC). Our new technique allows for the spatial predictions to be accompanied by a quantification of uncertainties in these predictions that reflect not only the coverage gaps but also the uncertainties in the MRLK parameters.

### 2.1.1 ABC posterior density estimation

Consider a $n$-dimensional spatial random variable $\mathbf{y}(\mathbf{x})$. The multi-resolution lattice kriging model depends on the unknown $p$-dimensional parameter $\boldsymbol{\theta}$. The probability distribution of the data given a specific parameter value $\boldsymbol{\theta}$ is denoted by $f(\mathbf{y}|\boldsymbol{\theta})$. If the prior distribution of $\boldsymbol{\theta}$ is denoted as $\pi(\boldsymbol{\theta})$, then the posterior density is given by

$$f(\boldsymbol{\theta}|\mathbf{y}) \propto f(\mathbf{y}|\boldsymbol{\theta}) \, \pi(\boldsymbol{\theta}) \tag{3}$$

Here, $\boldsymbol{\theta} = [\lambda, \ a.wght]^T$ where $\lambda$ and $a.wght$ are respectively the smoothing parameter and autoregressive weights (Nychka et al., 2019). These are the two main parameters of the MRLK. The autoregressive weight ($a.wght$) is the key covariance parameter. More precisely, it is the spatial autoregression (SAR) parameter that controls the spatial dependence among lattice points. It is essential for specifying and fitting the spatial model. The smoothness parameter $\lambda$ represents the signal to noise ratio: inappropriate estimate can lead to over or under fitting a spatial model and can result in imprecise interpolated values and prediction uncertainties (Nychka et al., 2015). The posterior distribution of these parameters given data, $f(\boldsymbol{\theta}|\mathbf{y})$, is approximated using ABC. The ABC acceptance-rejection technique based on variogram as a summary statistic is developed in the next section that is used to approximate the posterior densities.

### 2.1.2 Variogram-based ABC algorithm

Approximate Bayesian computation (e.g. Busetto and Buhmann, 2009; Beaumont, 2010; Dutta et al., 2017; Beaumont, 2019) is a family of algorithms that deals with the situations where the likelihood of a statistical model is intractable, whereas it is possible to simulate data from the model for a given parameter value. ABC bypasses the evaluation of the likelihood function by comparing observed and simulated data. Additionally, it offers algorithms that are very easy to parallelize. There are several forms of ABC algorithms. The standard rejection algorithm is the classical ABC sampler (e.g. Pritchard et al., 1999; Beaumont et al., 2002). It is widely used e.g. for model calibration by Gosling et al. (2018). The algorithm is based on drawing values of the parameters from the prior distribution. The data sets are simulated for each draw of parameters, each resulting in a chosen summary statistic. A distance metric is computed between the summary statistic of the observed and simulated data. The parameters that produce distances less than a tolerance threshold are retained. These accepted parameters form a sample from the approximate posterior distribution.

The basic idea of ABC is to simulate from the multi-resolution lattice kriging model for a given set of parameters $\boldsymbol{\theta}$. Simulations are run for a large number of parameters to be able to produce meaningful posterior distributions. The parameter values are retained for simulated data $\mathbf{y}^*$ that match the observed data $\mathbf{y}$ up to a tolerance threshold. For the similarity metric, we choose the sum of the squared differences between the semivariance at various lag distances of observed ($\gamma(h)$) and simulated ($\gamma^*(h)$) data. Indeed, these semivariances are traditional descriptors of the correlations across space.

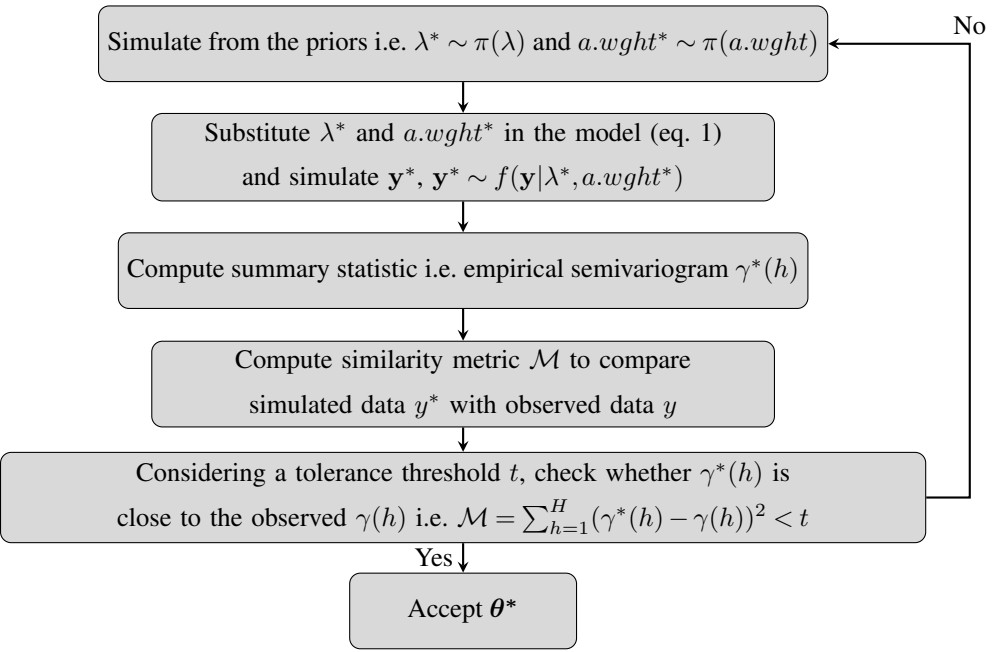

**Figure 1.** ABC acceptance-rejection algorithm using variogram as summary statistic for multi-resolution lattice kriging

Empirical semivariogram is typically computed by aggregating spatial pairs with similar distances. For this, the distances are partitioned into intervals called bins. The average distance of each bin is referred as lag distance $h$. Semivariance is half of the average squared differences between the observations for the point pairs falling in each bin. The standard rules are observed while computing the semivariogram. For example, the number of point pairs at each lag distance is at least 30.
Semivariogram is computed up to half of the maximum distance between the points over the whole spatial domain. The retained $\boldsymbol{\theta^*} = [\lambda^*, a.wght^*]^T$ are such that $\boldsymbol{\theta^*} \sim f_t(\boldsymbol{\theta}|\gamma)$.

## 2.2 Spatial estimate based on ABC posteriors

The existing MRLK inference methodology (Nychka et al., 2015) uses maximum likelihood estimators of smoothing parameter $\lambda$ and autoregressive weights $a.wght$. The ABC based inference methodology proposed here accounts for the posterior densities (not joint pointwise estimates) while making spatial predictions. For the full spatial estimate, the ABC posterior distributions of the MRLK model parameters ($\boldsymbol{\theta} = [\lambda, \text{a.wght}]^T$) are used to determine the conditional distribution of the coefficient $\mathbf{c}$ in (2). For the k$^{\text{th}}$ posterior sample,

$$[\mathbf{c}_k|\mathbf{y}, d, \rho, \theta_k] \sim N_m(\mu_{\mathbf{c}_k}, \boldsymbol{\Sigma}_{\mathbf{c}_k}) \tag{4}$$

Here, the mean ($d$) and process variance ($\rho$) are estimated using the maximum likelihood approach. The ABC posteriors of these parameters ($\rho$ and $d$) are not considered in this paper. Hence, the spatial predictions in ABC based multi-resolution lattice kriging are carried out using (5) and the associated uncertainties are evaluated based on (6) below:

$$\hat{\mathbf{y}} = \frac{\sum_{k=1}^{K}(\mathbf{1}_{(n)}\hat{d} + \boldsymbol{\Phi}\hat{\mu}_{\mathbf{c}_k})}{K} \tag{5}$$

$$\text{Var}(\hat{\mathbf{y}}) = \frac{\sum_{k=1}^{K}(\boldsymbol{\Phi}\hat{\boldsymbol{\Sigma}}_{\mathbf{c}_k}\boldsymbol{\Phi}^T)}{K} \tag{6}$$

## 3 HadCRUT4 data

The primary data used in this paper are the well-known HadCRUT4 (version 4.5.0.0) temperature anomalies (Morice et al., 2012). It is a combination of global land surface air temperature (CRUTEM4) (Jones et al., 2012) and sea surface monthly temperatures (HadSST3) (Kennedy et al., 2011a, b; Kennedy, 2014). The HadCRUT4 database consists of temperature anomalies with respect to the baseline (1961-1990). Monthly temperatures are provided beginning from 1850 over a $5° \times 5°$ grid. The average temperature anomalies of the stations falling within each grid are provided (Morice et al., 2012). The data set is updated on monthly basis to provide the updated climatic state. The gridded temperature estimates and time series can be downloaded from the Met Office website https://www.metoffice.gov.uk/hadobs/hadcrut4/. It is important to note that a newer and more refined version (HadCRUT5) of this data set is recently created (Morice et al., 2021). HadCRUT4 is not interpolated but the recently published HadCRUT5 is interpolated (Morice et al., 2012, 2021). Additionally, the compilation of the new Had-

CRUT5 data set involve a conditional simulation step that incorporates analysis uncertainties into an ensemble. Additionally, simulation is not involved in other data sets (Section 1) for calculating the uncertainty estimates in their interpolation.

## 3.1 Ensemble members

Errors in weather observations can either be random or systematic. They can lead to a complex spatial and temporal correlation
structures in the gridded data. An ensemble approach is used by Morice et al. (2012) to represent the observational uncertainties in HadCRUT4 data. The ensemble methodology characterizes the uncertainties that are spatially and temporally correlated. The realizations of an ensemble are typically formed by combining the observed data with multiple realizations drawn from the uncertainty model. This uncertainty model describes spatial and temporal interdependencies. This allows one-to-one blending of 100 realizations of HadSST3 and 100 realizations of CRUTEM4, resulting in 100 realizations of HadCRUT ensemble data.
Together these HadCRUT ensemble members represent the distribution of observational uncertainties that arise due to the non-climatic factors.

## 3.2 Observational uncertainties

Systematic observational errors emerge from the non-climatic factors. The HadCRUT ensemble data are created by blending the sea surface temperature anomalies from HadSST3 (Kennedy et al., 2011b, a; Kennedy, 2014) and land temperature anoma-
15 lies from CRUTEM4 (Jones et al., 2012). This approach follows the use of ensemble methodology to represent a range of observational uncertainties. HadSST3 is an ensemble data that is based on Rayner et al. (2006) uncertainty model. CRUTEM4 is not available as an ensemble data set (Jones et al., 2012). Therefore, it was converted to an ensemble data by Morice et al. (2012) using Brohan et al. (2006) uncertainty model.

The sea surface temperature anomalies are typically being measured using engine room intake measurements, bucket measurements, and drifting buoys. HadSST3 ensemble is used as the sea component of HadCRUT data. This ensemble is generated by drawing the bias-adjustment realizations for three measurement types. Therefore, the ensemble samples the systematic observational errors in sea surface temperature anomalies  (Rayner et al., 2006; Kennedy et al., 2011a; Kennedy, 2014). Below are the components of the error model used to characterize the observational uncertainties in the land measurements of HadCRUT.
This error model also generates ensemble version of CRUTEM4 (Morice et al., 2012).

    **Homogenization adjustment error**  Systematic biases occur due to changing station locations, measurement time, equipment and methods to calculate monthly averages. Homogenization adjustments are applied to the data to remove these non-climatic signals. These adjustments typically do not fully capture the systematic biases. This residual error is referred as homogenization adjustment error. It is modeled using Gaussian distribution (Brohan et al., 2006; Morice et al., 2012).

**Climatological error** For each station, temperature anomalies are computed with respect to the base period $1961 - 1990$. Typically, data are not available for all the months in the 30-year climatological period. These missing observations

introduce climatological error in the estimates of the base-period. The climatological error is modeled using a Gaussian distribution (Morice et al., 2012).

**Urbanization bias** The urban areas absorb and store more heat than the rural areas during the last few decades. This creates a heating effect that is known as the urban heat effect. The urbanization effect induces warming bias in the temperature records. The error model represents the effects of potential residual biases when using station records that have been screened for urbanisation. This bias is referred as urbanization bias. In HadCRUT, effects of urbanization are modeled on a global scale instead of considering these effects on measurement stations. For this, a truncated Gaussian distribution is used. The large-scale urbanization bias in temperatures is adjusted for all the years beyond 1900 (Morice et al., 2012).

**Exposure bias** It has been observed that bias in temperatures can be introduced due to the station siting and exposure. Changes in instrumentation can broadly be grouped into two broad classes. There were few standards for thermometer exposure or instrument shelters before the 19th century. By the early 20th century, these thatched (or covered) enclosures were largely replaced by free-standing louvered shelters or Stevenson type screens (Trewin, 2010). A Stevenson type screen is a shelter or enclosure that protects meteorological instruments from precipitation and direct heat radiation. However, it allows free circulation of the air. Changes in the thermometers, exposure to the atmosphere and shelters from direct or indirect solar radiation introduces exposure bias in temperatures (Parker, 1994; Moberg et al., 2003). This error in temperatures is modeled on a regional scale in HadCRUT using a Gaussian distribution (Morice et al., 2012).

In addition to the large-scale bias terms that are discussed above, Morice et al. (2012) provides the measurement and grid sampling uncertainty components. These are particularly important for marine regions as ship/buoy movement leads to spatially correlated errors. HadCRUT4 does not include these in the ensemble. These are provided as additional spatial error covariance matrices. The data set (Section 4) created in this paper uses HadCRUT4 ensembles only.

## 4  Hyperparameter temperature ensemble data

The ensemble temperature data set created by Ilyas et al. (2017) presumed perfect knowledge of multi-resolution lattice kriging covariance parameters. The approximate Bayesian computation based multi-resolution lattice kriging developed in Section 2.1 is applied to the sparse HadCRUT4 ensemble data (Section 3, Morice et al. (2012)). As a result of this, a new 100,000 member ensemble data is created. It is an update to the data set discussed in Ilyas et al. (2017).

The key difference between the two data sets is the inference methodology. The updated data set is produced by using the ABC based posterior densities of the multi-resolution lattice kriging covariance parameters whereas the first data set used pointwise estimates obtained via a likelihood approach. The use of posterior distribution of the model parameters creates a data set that accounts for the multi-resolution lattice kriging parametric uncertainties.

## 4.1 ABC posteriors and model parameters

The HadCRUT ensemble data set samples observational uncertainties in the instrumental temperature records (Morice et al., 2012). Similar to the first version of Ilyas et al. (2017), the updated data set is based on HadCRUT4 ensemble members. For the updated version, the ABC posterior densities of the smoothing parameter and autoregressive weights are determined. These are identified for each of the 2028 months from January 1850 to December 2018. The ABC algorithm (Figure 1) based on the variogram is used to compute the posterior distributions. Uniform priors are considered i.e. $U(0.001, 4)$ and U(1, 4) for the smoothing parameter and autoregressive weight, respectively. In particular, priors put on autoregressive weight assume that omega (Nychka et al., 2019) follows a uniform distribution over the interval [ -4.5, .55]. This choice covers a useful span of spatial correlations when omega is translated back into the $a.wght$ (i.e. $a.wght = 1 + exp^{2*\text{omega}}$) parameter and subsequently into the dependence of the field at the lattice points. The tolerance threshold $t$ is chosen to correspond to the $4\%$ acceptance rate with 250 iterations. It results in 10 hyperparameter sample draws from the posterior densities.

Fitting the multi-resolution lattice kriging model requires a choice of the basis functions and marginal spatial variance. The multi-resolution basis is the same as the one that was chosen for the earlier version of the ensemble (Ilyas et al., 2017). So a three-level model is chosen such that the number of basis function is greater than the number of spatial locations. The value of $\boldsymbol{\alpha}$ i.e. the marginal spatial variance (Nychka et al., 2015) is estimated as $\boldsymbol{\alpha} = (0.2451, 0.01606, 0.7389)^T$. This is computed over the field with the maximum available information i.e. February 1988. This slightly varies across time. The values of $\boldsymbol{\alpha}$ parameter for February 1988 are the relative variances between different resolution levels. The decay of these for finer resolution fix the smoothness of the field and because these are difficult to estimate they were not emphasized in the analysis and the correlation range parameter, $a.wght$, was given more attention. February 1988 has the most complete data field. Therefore, it was assumed that the smoothness reflected in the $\boldsymbol{\alpha}$ parameters are consistent across different times. It results from the maximum likelihood estimation as the algorithm in Figure 1 is not yet extended to obtain posteriors of $\boldsymbol{\alpha}$ as this parameter has little influence on the uncertainties compared to the others. Other parameters are estimated for each monthly field since the spatial characteristics can vary considerably. The geodesic grid and the great circle distance is used to handle the spherical domain. To implement multi-resolution lattice kriging, the LatticeKrig R package version 6.4 is used. As an example, the posterior densities of smoothing parameter $\lambda$ and autoregressive weight $a.wght$ are shown in Figure 2 for one spatial field. Other two examples are presented in the appendix (Figure C1 and Figure C2). These posterior distributions result from the HadCRUT4 median spatial field with the minimum spatial coverage i.e. May 1861. The month with the poorest spatial coverage was chosen to illustrate the advantages (Figure 3 and Figure 4) of using a more refined spatial approach (Section 2.1). It is important to note that hyperparameter estimates are global. These are estimated independently for each field without using any regional estimates. Additionally, temperature anomaly variability is assumed identical at all locations over land and sea. Also, it is a space only model (not space-time). There is no accounting for the persistence of temperatures used to aid reconstruction or accounted for in uncertainty estimates.

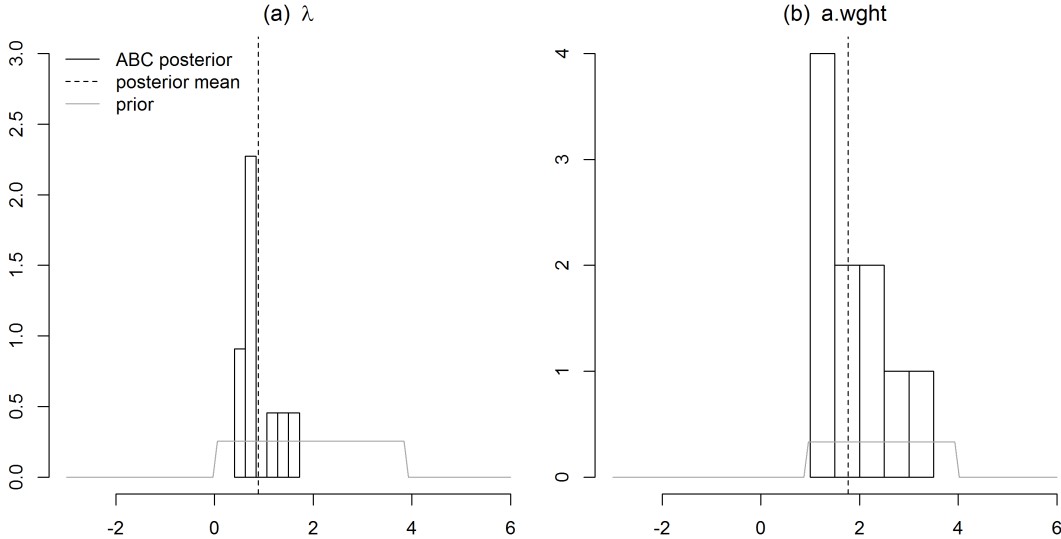

**Figure 2.** Posterior densities of (a) the smoothing parameter, $\lambda$, and (b) the autoregressive weight, $a.wght$ for HadCRUT4 spatial field with the minimum spatial coverage i.e. May 1861.

## 4.2 Spatial field with minimum coverage

ABC-based multi-resolution lattice kriging (Figure 1) is used to predict this sparse spatial field. The spatial predictions and associated uncertainties are shown in Figure 3a and Figure 4a, respectively. The spatial predictions (Figure 3a) are computed using the available spatial observations, multi-resolution basis functions (Section 4.1) and ABC posterior distributions of au-

5    toregressive weights $a.wght$ and smoothing parameter $\lambda$ (Figure 2). Equation (5) is used for the calculation of the spatial predictor. For comparison with the previous reconstruction (Ilyas et al., 2017) of this spatial field, the spatial predictions based on profile maximum likelihood approach are presented in Figure 3b.

The value of autoregressive weight $a.wght$ used in the old ensemble is $8.95$. This value is much greater than the range of

10   the posterior distribution of $a.wght$ (Figure  2). This is due to the fact that the version of LatticeKrig package used here and in old ensemble (Ilyas et al., 2017) are 6.4 and 6.2, respectively. LatticeKrig version 6.4 is mainly an update on the LKrig function for spherical spatial domains. The minimum value of autoregressive weights in version 6.2 is restricted to be greater than 6 to avoid artifacts in the covariance. This restriction is removed in version 6.4 and the weights are updated for improved specification of covariance over the sphere (Nychka et al., 2019).

The difference of these reconstructions (Figure 3c) indicate that the spatial predictions using ABC-based multi-resolution lattice kriging (Figure 3a) generally show anomalies in temperature from the baseline climate that are smaller in magnitude compared to the likelihood-based reconstruction (Figure 3b). The uncertainties in predictions are shown in Figure 4a that

result from ABC-based multi-resolution lattice kriging. These uncertainties in the predictions are computed using equation (6). Figure 4c compares these uncertainties with those resulting from the previous reconstruction (Figure 4b). It can be observed that ABC based multi-resolution lattice kriging is producing higher uncertainty estimates close to the observed spatial sites. This was expected since there is now account of more sources of uncertainty. However, the unobserved grid locations are showing less uncertainties that are resulting from ABC based multi-resolution lattice kriging.

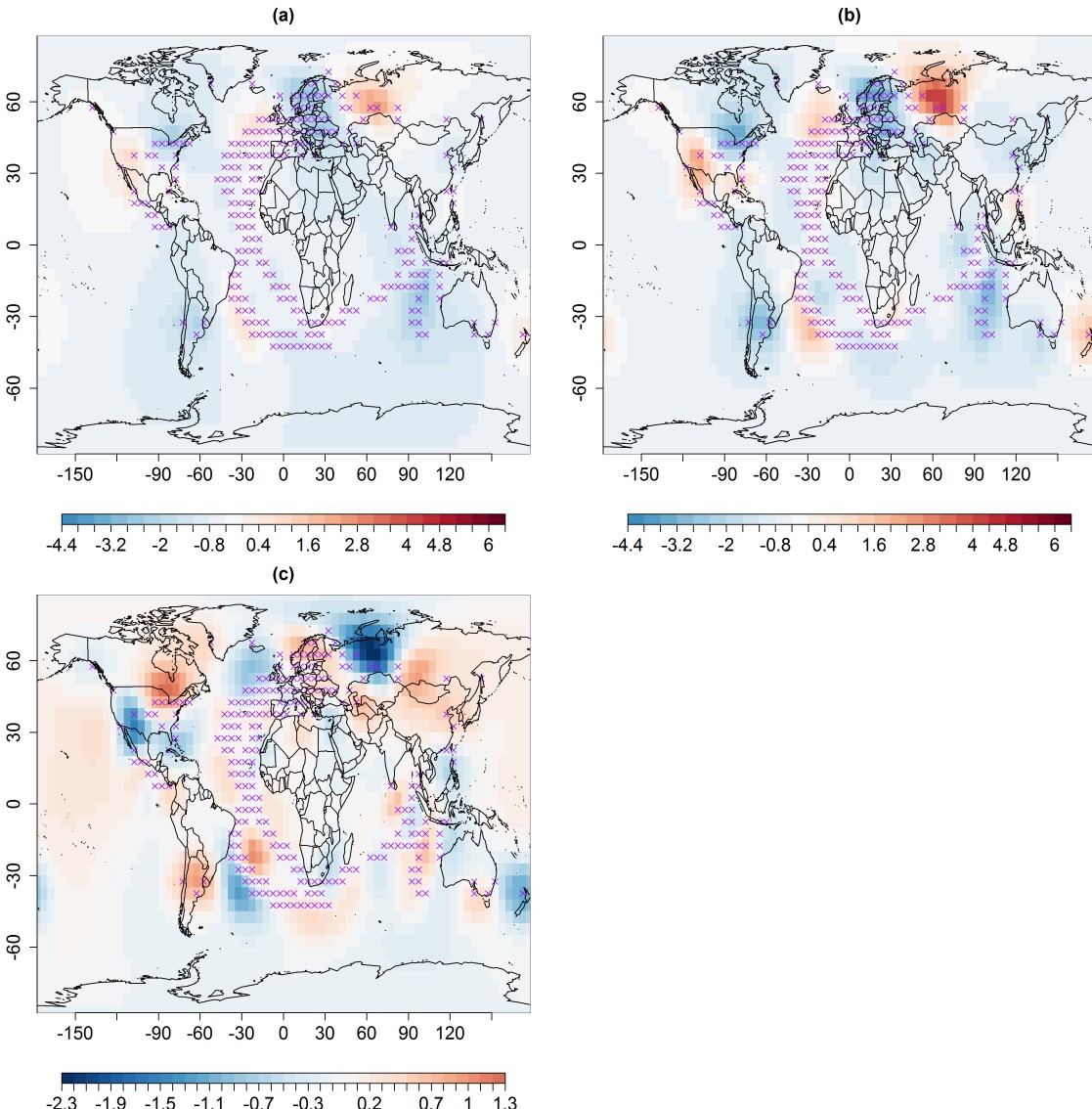

**Figure 3.** Median spatial predictions (in °C) for May 1861 using multi-resolution lattice kriging based on (a) ABC using variogram (Section 2.1) and (b) profile maximum likelihood approach (Nychka et al., 2015) used to create data in Ilyas et al. (2017). (c) Difference of (a) and (b) i.e. (a)-(b). × signs show observed spatial sites (purple).

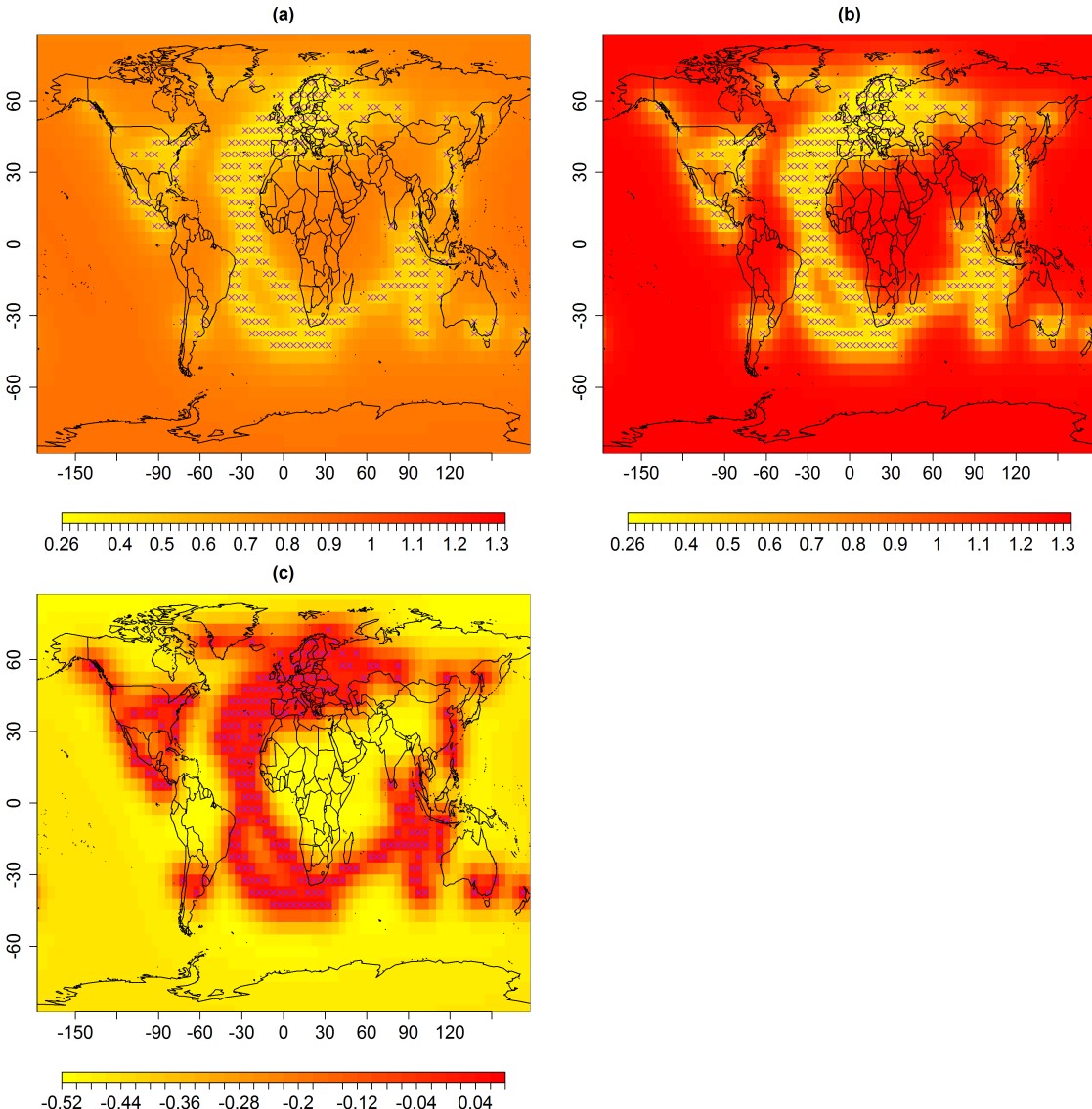

**Figure 4.** Standard error (uncertainties) in °C associated with spatial predictions for median May 1861 using multi-resolution lattice kriging based on (a) ABC using variogram (Section 2.1) and (b) profile maximum likelihood approach (Nychka et al., 2015) used to create data in Ilyas et al. (2017). (c) Difference of (a) and (b) i.e. (a)-(b). × signs show observed spatial sites (purple).

It seems like the ABC based multi-resolution lattice kriging is leaning towards more of the observational noise variance and less process variance. Additionally, possibly it is due to the fact that LKrig function of LatticeKrig R-package has undergone substantial modifications. The spatial estimates and uncertainties of two more fields are presented in Figure B1 to Figure B4.

These fields generally show a similar story as seen in the May 1861 – the ABC-based predictions have collapsed to be more certain about a weaker temperature anomaly than seen in the Ilyas et al. (2017) dataset.

## 4.3   100,000 member hyperparameter ensemble

The ABC posterior distributions and model parameters of the multiresolution lattice kriging model (Section 4.1) are used to generate an ensemble. This ensemble data is based on HadCRUT4 temperature data. The HadCRUT4 monthly data set consists of 100 sparse ensemble members. For each of 100 monthly spatial fields of HadCRUT4, a spatially complete 1000 member ensemble is created that samples the coverage and parametric uncertainties of multi-resolution lattice kriging. The resulting 100,000 ensemble members are referred as a hyperparameter temperature ensemble data set. The 1000 members of ensemble generated from each of the 100 HadCRUT4 ensemble members (thus eventually creating $1000 \times 100 = 100,000$ fields) are the random fields drawn from the multivariate conditional normal distribution. These are drawn by conditioning on the HadCRUT4 available field measurements, and sampling the multi-resolution lattice kriging covariance model, namely the variogram based ABC posteriors of autoregressive weights and smoothing parameter. In other words, 100 fields are drawn from the multivariate conditional normal distribution. These are sampled corresponding to each of the 10 draws from the ABC posterior distributions of smoothing parameter and autoregressive weights.

This ensemble data set is generated using High-Performance Computing due to the computational expense. The HadCRUT4 sparse monthly data set spans from 1850 to 2018, 2028 months in total. For each month, the posterior of autoregressive weight $a.wght$ and smoothing parameter $\lambda$ are computed using the median ensemble member. Given these posteriors, 1000 coverage samples are drawn for each of 100 HadCRUT4 ensemble members. To achieve sufficiently fast computation, different parts of the data are handled in parallel on different nodes. For this, 100 HadCRUT4 ensemble members are divided into sets of 5. It results in 20 sets each consisting of 5 members. For one time point, the sampling for 100 HadCRUT4 ensemble members is performed in parallel by submitting 20 shared memory parallel jobs with 5 threads. Therefore, a single job that performs computations over 5 ensemble members of a month runs in parallel and takes approximately 66 minutes. The total number of parallel jobs are $20 \times 2028 = 40560$. Therefore, the time required to run these jobs is $40560 \times 66$ minutes $= 2676960$ minutes $= 61.97$ months. Typically, 6 or 7 parallel jobs run simultaneously so it took approximately eight months of wall-clock time to perform these computations.

## 5   Uncertainties in global mean temperature

The global mean temperature time series are computed for the 100,000 member hyperparameter ensemble data described in Section 4.3. For each ensemble member, the global mean time series is calculated. Figure 5 represents the annual median time series along with the $95\%$ credible interval. For comparison, Figure 5a also presents the median time series and the uncertainties resulting from an earlier version of the ensemble data (Ilyas et al., 2017), which sampled only the combination of observational and coverage uncertainties without uncertainty in the MRLK model.

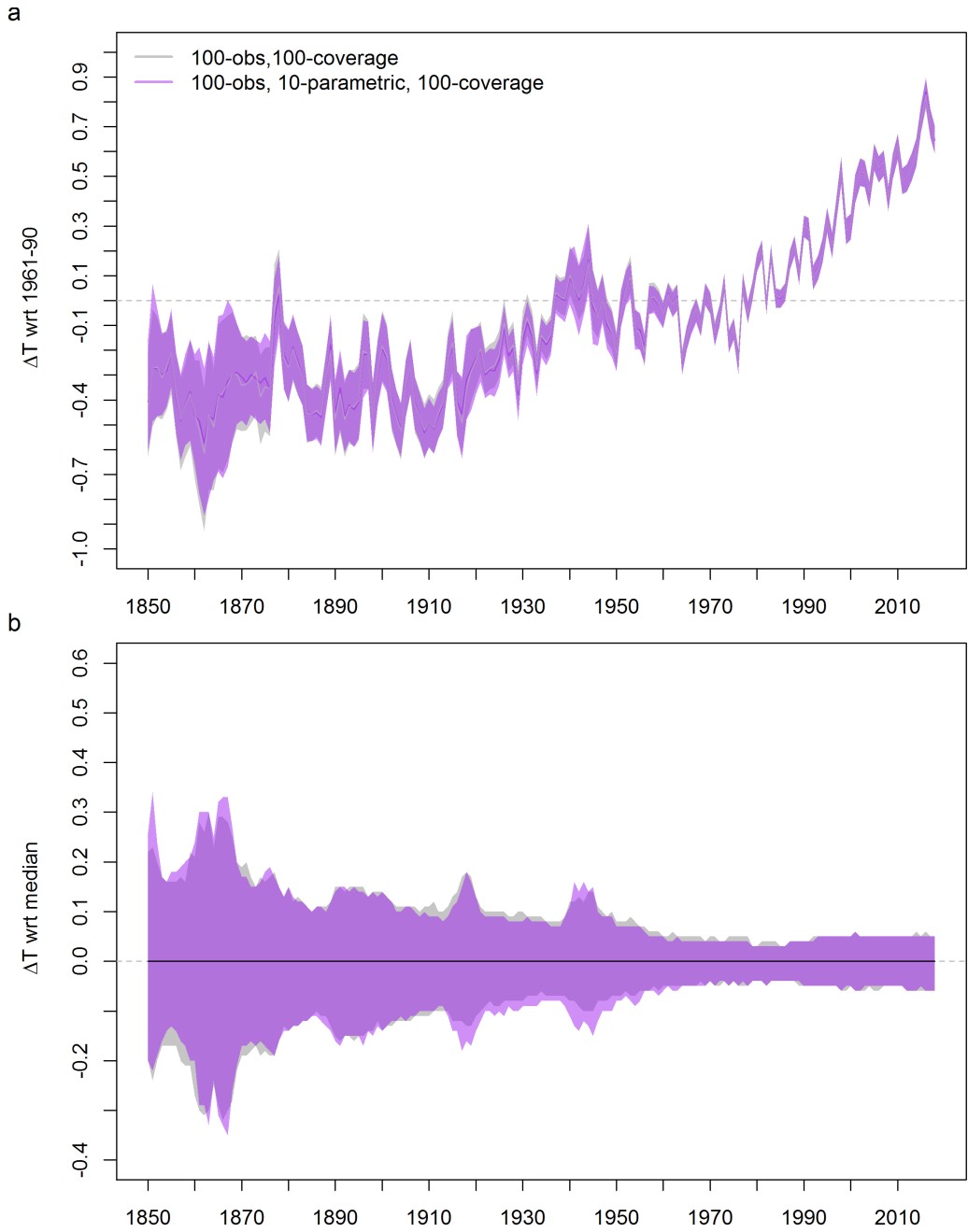

**Figure 5.** Global mean, annual average temperature anomalies in °C with respect to a) 1961-90 baseline and b) median. 95% credible (purple) and confidence (grey) interval estimates based on the data set created in Ilyas et al. (2017) and hyperparameter ensemble and the data set created in Section 4.3.

The impact of including parametric uncertainties can be observed from figure 5 at a global scale. The overall features of the time series resulting from the hyperparameter ensemble and the first version of data (Section 4.3) are mostly similar (Figure 5b). Uncertainty ranges appear to be roughly comparable, of similar magnitude to the quantisation in the plot of roughly $0.01°C$, but slightly skewed one way or another. The smoothing parameter $\lambda$ is process to noise ratio (Nychka et al., 2015). Sampling into high values of smoothing parameter can give a process with low variance and large measurement noise, which can lead to smaller uncertainties arising from sampling limitations. The lower variance of the ABC analysis field (Figure 3) suggests that this might be the case.

## 6  Subsample of hyperparameter ensemble data

For easy handling of this large data, a subsample of this hyperparameter ensemble is created using Conditioned Latin Hypercube Sampling (CLHS) (Minasny and McBratney, 2006). In practice, Monte Carlo and Latin Hypercube sampling approaches are used to draw samples that approximate the underlying distribution. Usually, a large number of samples are required to achieve good accuracy in traditional Monte Carlo (e.g. Pebesma and Heuvelink, 1999; Olsson and Sandberg, 2002; Olsson et al., 2003; Diermanse et al., 2016). Additionally, the Monte Carlo samples can contain some points clustered closely while other intervals within the space get no sample. On the other hand, the Latin hypercube sampling provides a stratified sampling framework for improved coverage of the k-dimensional input space (e.g. McKay et al., 2000; Helton and Davis, 2003; Iman, 2008; Clifford et al., 2014; Shields and Zhang, 2016; Shang et al., 2020). Conditioned Latin hypercube sampling is an attempt to draw a sample that captures the variation of multiple environmental variables. This sample accurately represents the distribution of the environmental variables over the full range.

To draw a subsample from 100,000 ensemble members, we considered a set of prominent environmental variables i.e. monthly area averages and IPCC AR5 regional means. The conditional Latin hypercube sample is being drawn from the distribution of these environmental variables. The subsample accurately approximates the variation of the set of environmental variables over the full range of these variables. Stating differently, the distributions of the set of environmental variables in the conditioned Latin hypercube sample of size 100 is approximately similar to the distributions of these variables over the full range based on 100,000 ensemble members. As an example, the distribution of a grid box is shown in Figure 6 for May 1861. The full ensemble distribution is based on 100,000 gridboxes (Section-4.3). The subsample distribution results from the conditioned Latin hypercube subsample of 100 gridboxes. Both the distributions overlap mostly. However, the extreme values at the tails are not being captured by the subsample. Also, It is important to note that the subsample ensemble only captures the variation of the specified environmental variables discussed above (i.e. AR5 regional means and monthly area averages). This subsample may not be suitable to explore any other locations outside of those regions. In those situations, it might be wise to perform a check using the full hyperparameter ensemble.

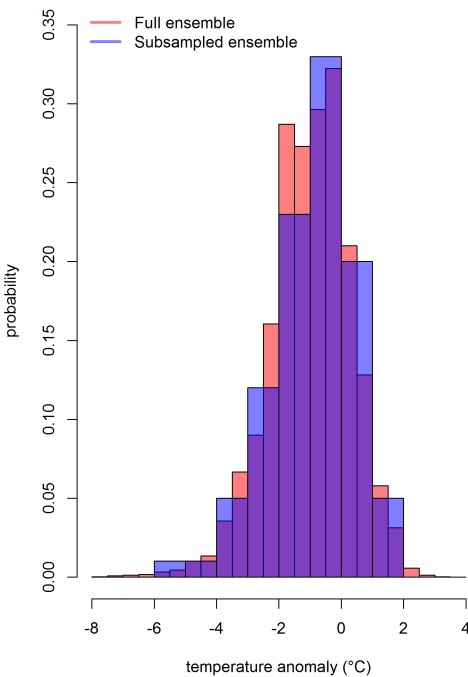

**Figure 6.** Distribution of the grid box, (lon, lat) = (72.5°, 32.5°) that includes Lahore for May-1861. Full ensemble consists of 100,000 gridboxes (Section- 4.3) and subsampled ensemble consists of 100 gridboxes based on Latin Hypercube sampling.

## 7 Discussion and Conclusion

Uncertainty in gridded temperature comes from a variety of sources, of which instrumental error is only one. Uncertainties associated with the lack of spatial coverage are understandably more important in the early portion of the record – when observations were sparse. Many approaches can be used to fill in the grid boxes missing observations, some of which also quantify

5 the associated coverage uncertainties. There is debate as to which model is the most appropriate. For the first time, this research considers the uncertainty in that model itself. We demonstrate that this is achievable through Monte Carlo sampling of perturbations in the 'hyperparameters' in the model. The method described is computationally intensive and results in a dataset that is somewhat unwieldy. We have demonstrated that this latter point can be overcome by conditional sampling under some circumstances.

The hyperparameter ensemble (i.e. an update on Ilyas et al. (2017) ) has provided an improved version of the global temperature anomalies since 1850. Instead of classical or frequentist approach, a Bayesian methodology for quantification of uncertainties in large data settings is developed for characterization of uncertainties. The impact of including parametric uncertainties is evident at a regional (Figure 4) and global scale (Figure 5). However, the overall impact of parametric uncertainties

15 makes little substantive advance in our general understanding of global temperature estimates since 1850. The hyperparameter

sampling approach described here results in an ensemble that is an order of magnitude larger than that of Ilyas et al. (2017) and three orders of magnitude larger than the original HadCRUT4 ensemble of Morice et al. (2012). Our analysis here has focused on changes in the mean climate, and perhaps this hyperparameter ensemble may be better suited for studies that aim to explore changing climate variability (Beguería et al., 2016).

Given the small differences in the resultant temperature reconstructions, it is not clear that additional effort to create a hyperparameter ensemble is justified. We strongly recommend that before embarking on future efforts people explore this dataset to see if there will be any tangible benefit. It may be possible that alternate methodologies could be devised that do not require such outlay of resources during creation and analysis. For example, there may be approaches to sample hyperparameters and uncertainty mappings simultaneously, without running the risk of undersampling key elements of parameter space. Or a few example fields could be created using the approach here, then conditional latin hypercube sampling used to determine the hyperparameter settings for a reduced ensemble.

*Acknowledgements.* This research was supported by a UCL Overseas Research Scholarship (M.I.), a University of the Punjab Overseas Scholarship (M.I.) and Research Grant (M.I), and National Center for Atmospheric Research (NCAR) (M.I.). This research would not have been possible without the assistance and generosity of the Met. Office's Hadley Centre and the University of East Anglia's Climate Research Unit. The authors acknowledge the use of the UCL Legion High Performance Computing Facility (Legion@UCL), and associated support services, in the completion of this work. We also acknowledge helpful comments from the reviewers.

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

# Appendix A

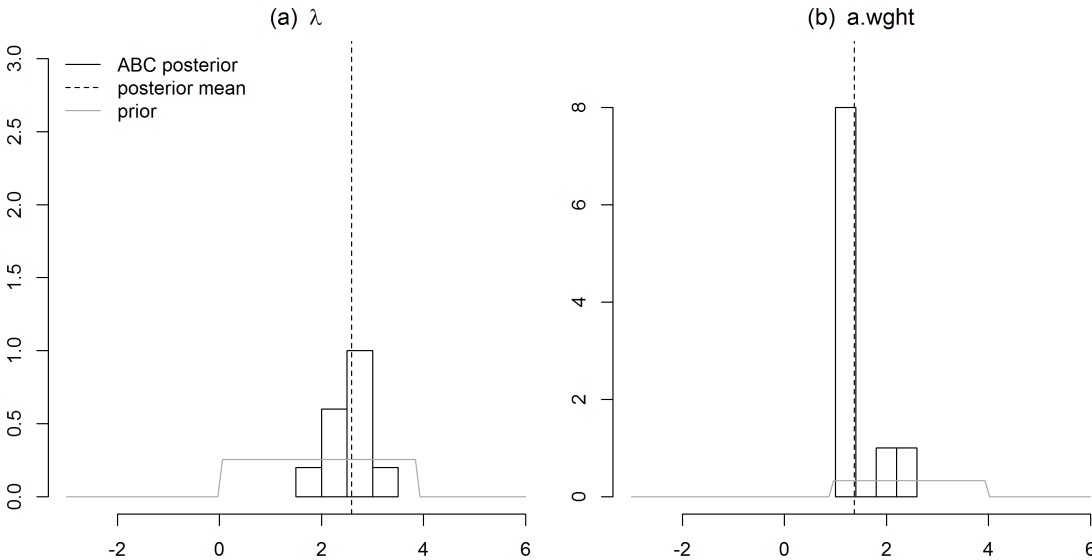

**Figure A1.** Posterior densities of smoothing parameter $\lambda$ (a) and autoregressive weight, a.wght (b) for HadCRUT4 spatial field with the $50\%$ spatial coverage i.e. February 1932.

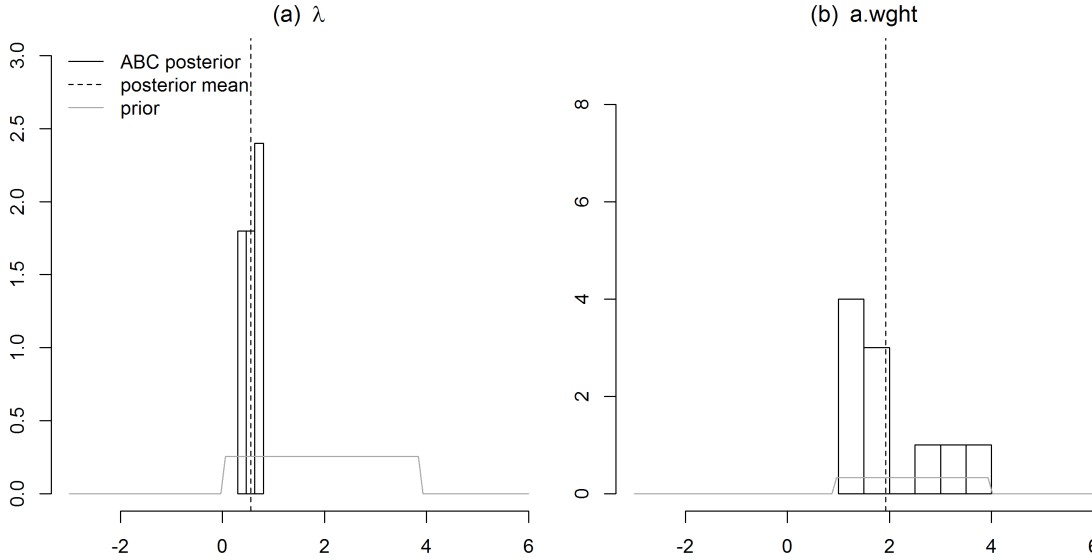

**Figure A2.** Posterior densities of smoothing parameter $\lambda$ (a) and autoregressive weight, a.wght (b) for HadCRUT4 spatial field with the maximum ($78\%$) spatial coverage i.e. February 1988.

## Appendix B

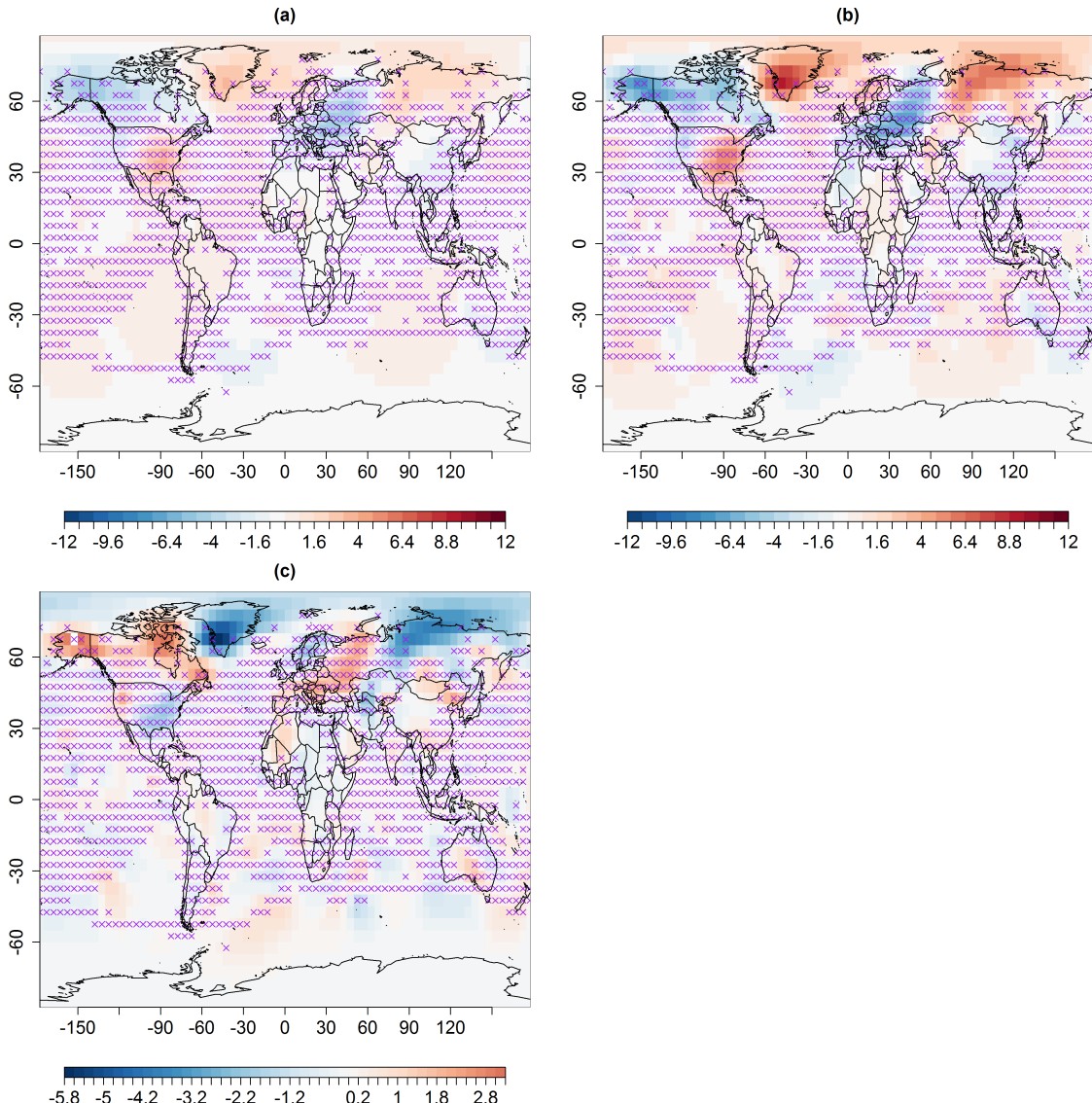

**Figure B1.** Spatial predictions (in °C) for February 1932 using multi-resolution lattice kriging based on (a) ABC using variogram (Section 2.1) and (b) profile maximum likelihood approach (Nychka et al., 2015) used to create data in Ilyas et al. (2017). (c) Difference of (a) and (b) i.e. (a)-(b). × signs show observed spatial sites (purple).

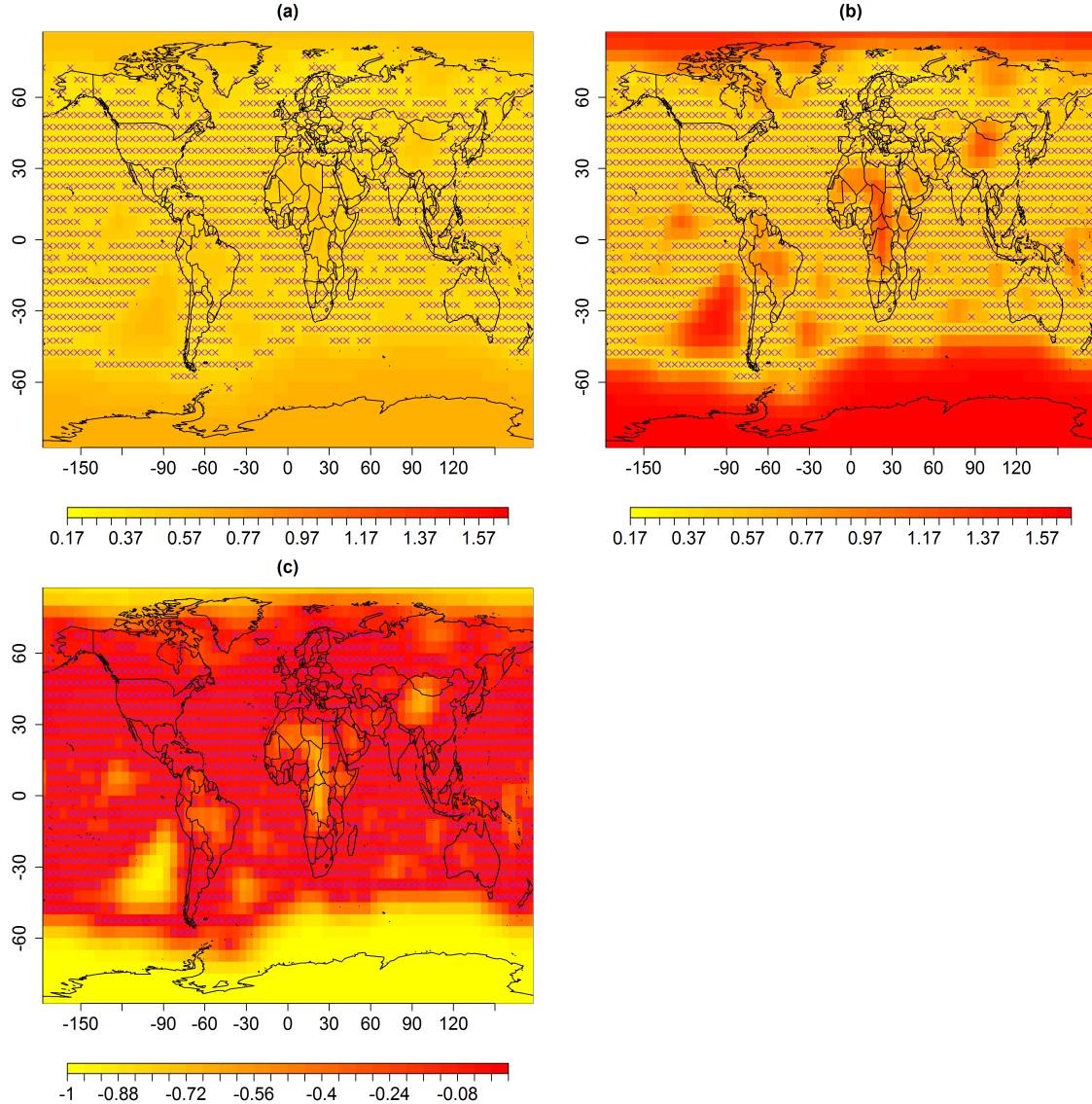

**Figure B2.** Standard error (uncertainties) in °C associated with spatial predictions for median February 1932 using multi-resolution lattice kriging based on (a) ABC using variogram (Section 2.1) and (b) profile maximum likelihood approach (Nychka et al., 2015) used to create data in Ilyas et al. (2017). (c) Difference of (a) and (b) i.e. (a)-(b). × signs show observed spatial sites (purple).

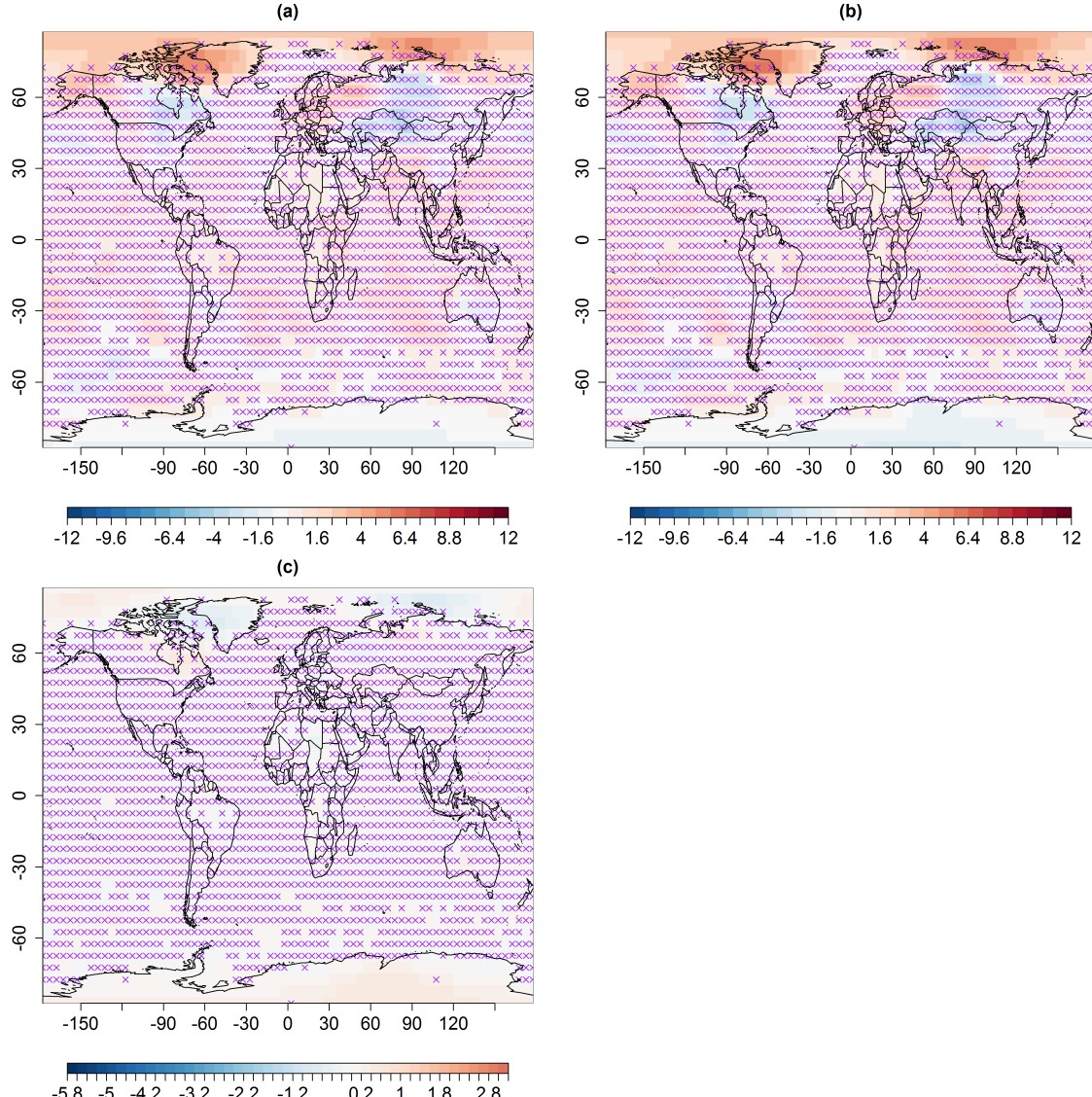

**Figure B3.** Spatial predictions (in °C) for February 1988 using multi-resolution lattice kriging based on (a) ABC using variogram (Section 2.1) and (b) profile maximum likelihood approach (Nychka et al., 2015) used to create data in Ilyas et al. (2017). (c) Difference of (a) and (b) i.e. (a)-(b). × signs show observed spatial sites (purple).

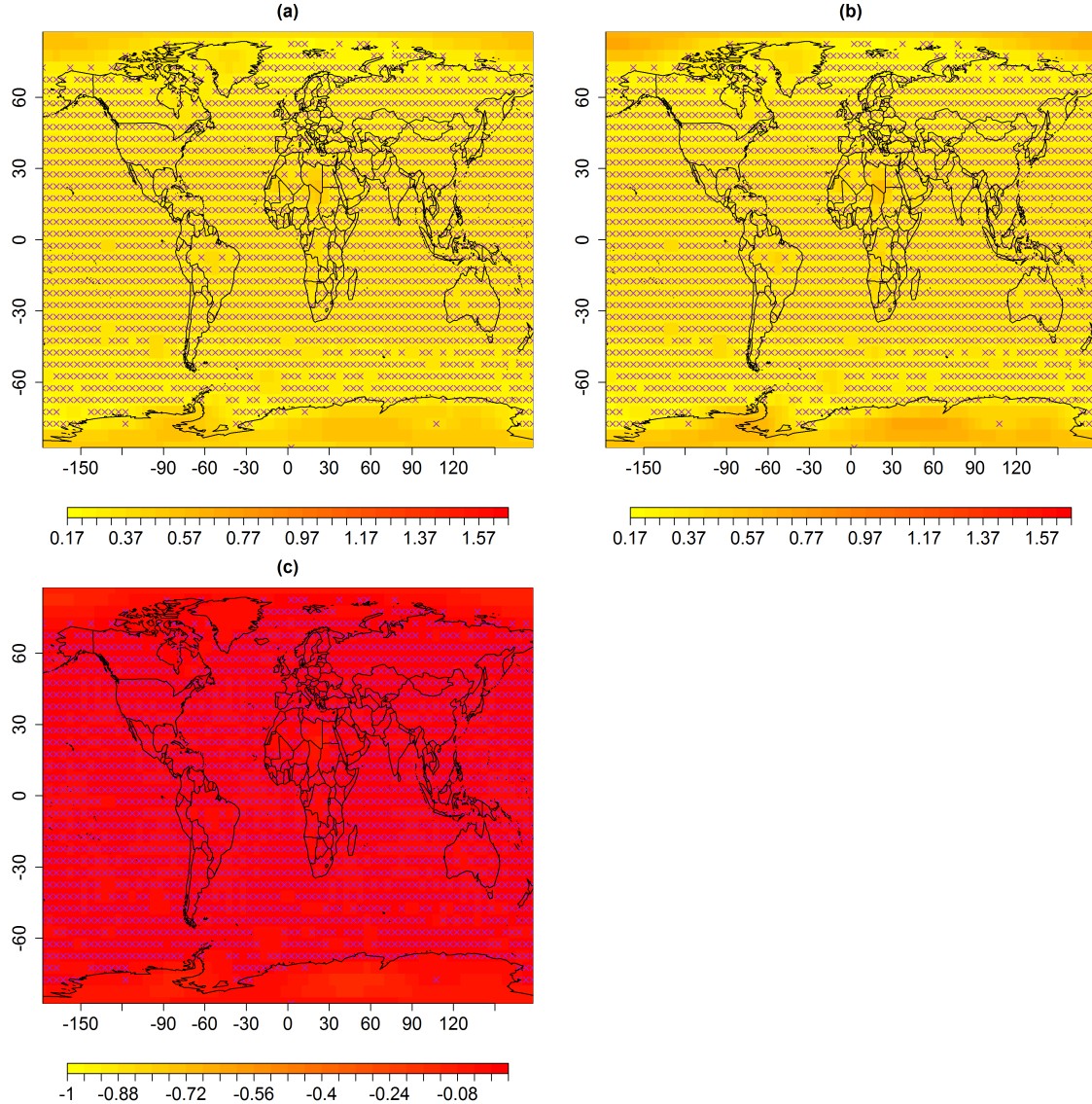

**Figure B4.** Standard error (uncertainties) in °C associated with spatial predictions for median February 1988 using multi-resolution lattice kriging based on (a) ABC using variogram (Section 2.1) and (b) profile maximum likelihood approach (Nychka et al., 2015) used to create data in Ilyas et al. (2017). (c) Difference of (a) and (b) i.e. (a)-(b). × signs show observed spatial sites (purple).

**Appendix C**

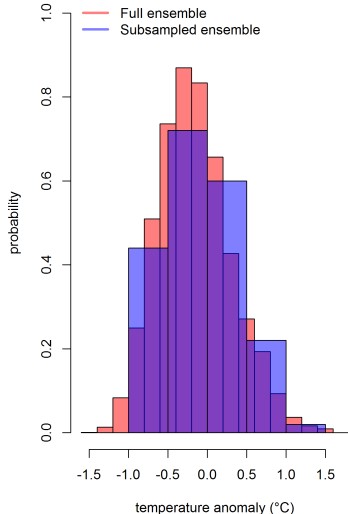

**Figure C1.** Distribution of the grid box, (lon, lat) = (72.5°, 32.5°) that includes Lahore for Feb.-1932. Full ensemble consists of 100,000 gridboxes (Section- 4.3) and subsampled ensemble consists of 100 gridboxes based on Latin Hypercube sampling.

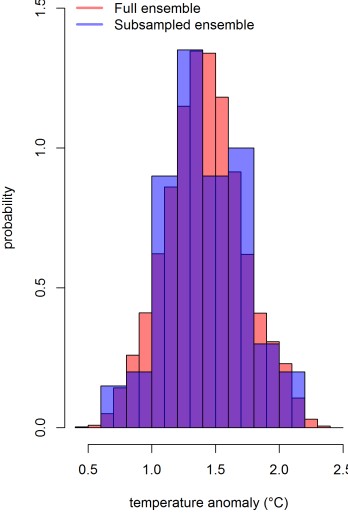

**Figure C2.** Distribution of the grid box, (lon, lat) = (72.5°, 32.5°) that includes Lahore for Feb.-1988. Full ensemble consists of 100,000 gridboxes (Section- 4.3) and subsampled ensemble consists of 100 gridboxes based on Latin Hypercube sampling.