# Peer review of "Global Ensemble of Temperatures over 1850-2018: Quantification of Uncertainties in Observations, Coverage, and Spatial modelling (GETQUOCS)"

_Atmospheric Measurement Techniques, 2020_

## Author Response (AR1)

**Response to reviews for revised paper**

**Referee-1:**

Comment: An interesting paper with a few issues to be resolved. In their analysis of global temperature data using the multiresolution lattice kriging method, the authors extend the work of Ilyas et al 2017 by exploring the hyperparameter estimation uncertainty using a Monte Carlo sampling method. It is interesting to see the impacts of this hyperparameter uncertainty assessment. These are a potentially important source of uncertainty in assessments of observed global temperature change that have not previously been investigated in other studies. There are some issues with the paper structure, including a lack of concluding remarks. Additional discussion of the effects controlled by the sampled parameters and illustration of the impacts of their sampling throughout the temperature record is needed.

Response: We are grateful for the encouraging comments. Section-6 discussion is replaced by conclusion and discussion where concluding remarks are added.

Comment-1: While it is great that the paper includes estimates of hyperparameter uncertainty, I'm am left uncertain on the extent to which hyperparameter uncertainty translates into an appreciable uncertainty in the temperature fields and how this varies through the temperature record. The paper only provides examples of hyperparameter estimates and resulting fields for a single month. Is this representative of other months? Interpretation of differences between the ABC based and profile likelihood based analyses would be aided by showing how the profile likelihood hyperparameter estimates compare to those from ABC, for example in Figure -2. Similarly, temperature/uncertainty fields are only shown for a single month in Figures 3/4.

These points seem important to understand the benefits of sampling the hyperparameter uncertainty, also given that the method appears to be computationally expensive.

Response: Uncertainties in the spatial field pretty much depends upon the spatial coverage. This single field was chosen to highlight the maximum impact of parametric uncertainties as it has the least spatial coverage. The paper now includes spatial estimates and their corresponding standard errors in the appendix for two more time points. Section 4.2, second paragraph discusses the hyperparameter estimates and profile likelihood estimates.

Comment-2: The review of prior literature is frequently a few years out of date and needs updating. Some cited studies are inaccurately or incorrectly described. See detailed comments for details.

Response: Literature of more recent related studies is added. Citations mistakes are corrected as well.

Comment-3: There appear to be a few simplifications in the statistical model/uncertainty model used that are not discussed:

Ø Hyperparameter estimates are global, estimated independently for each field, with no regional estimates, essentially modelling temperature anomaly variability as being identical at all locations over land and sea.

Ø As a space only model (not space-time) there appears to be no accounting for persistence of temperatures used to aid reconstruction or accounted for in uncertainty estimates.

Ø My understanding of MRLK is that it models observational error as identically distributed for each observed location. The analysis described makes no mention of the additional uncertainty terms for HadCRUT4 (in addition to the ensemble) that describe differences in observational error distributions for each grid cell and correlations in errors between grid cells, arising from the movement of marine measurement platforms. It appears that this information, that is not encoded into the HadCRUT4 ensemble members, has not been used and they are not described in Section 3.2. These uncertainties were found to be important in Morice et al 2012. Some comment on not including these, or how they are approximated by the additive uncorrelated error term in MRLK, would be appropriate.

Response: Apologies for insufficiently describing the statistical/ uncertainty model. Section 4.1 and Section 3.2 discusses these points now.

Comment-4: The reader needs to refer to Nychka et al 2015 to understand the meaning of the lambda hyperparameter. The "aw" hyperparameter (please rename to use a single letter unless it is a product of two variables) does not appear to be described in Nychka et al 2015. I do not understand how to interpret the function of this "autoregressive weights" hyperparameter and how it might affect the resulting temperature fields.

Response: Apologies for this confusion. We have replaced "aw" by a single letter "a.wght". This notation is used in LatticeKrig R-package. Autoregressive weight is a sort of range parameter that is used in the precision matrix of Gaussian Markov random field. This is defined in section 2.1.1.

Comment-5: 5. Discussion of Figure 5 suggests that uncertainty estimates for global average temperature anomalies are wider for ABC that those of Ilyas et al 2015, but this is not particularly evident in Figure 5. Uncertainty ranges appear to be roughly comparable, of similar magnitude to the quantisation in the plot of roughly 0.01 °C, but slightly skewed one way or another. It's not clear that the ABC sampling of the hyperparameter uncertainty would necessarily lead to wider uncertainty estimates in the global mean than the profile likelihood estimates. For example, the lambda parameter is defined in Nychka et al 2015 as lambda =noise variance / process variance. Sampling into high values of lambda would give a process with low variance and large measurement noise, which would lead to smaller uncertainties arising from sampling limitations. The lower variance of the ABC analysis field in Figure 3 suggests that this might be the case. It would be an alternative explanation to the changes in LatticeKrig 6.4 that are alluded to in the first paragraph on page 10.

Response: Thanks for the alternative explanation. This is added in section-5.

Comment-6: The paper ends rather abruptly with a discussion of a sampling method (which arguably should be moved earlier in the paper). It would benefit from a conclusions section. Are there any deficiencies in the approach that we should be aware of? What's missing or could further developed? It could comment on developments while this paper was being worked on that are not included, e.g. for HadSST4 and talk more broadly about where this study fits alongside other research in the subject area. It would be an appropriate place to place a link to the data.

Response: Conclusion and discussion section is improved. Latin hypercube details are moved earlier.

**Detailed points.**

Comment: Page 1, Abstract, line 1: Needs the word global in there to indicate that we're talking about global temperature records?

Response: Done.

Comment: Page 1, Abstract, Line 7: It's not clear in the abstract what the "variation" in parameters is referring to. Hyperparameter estimates vary from month to month but not spatially. Or is this referring to the uncertainty in hyperparameter values, which otherwise isn't stated in the abstract and is the key addition in the paper?

Response: Done.

Comment: Page 1, line 15 – Good et al 2016 is a satellite-based skin temperature record, not air temperature?

Response: This reference is removed.

Comment: Page 1, line 19 – It's not exactly so simple as obtaining data from the WMO/GCOS. Modern messages are transmitted via these means but much work is required to compile observations from individual nation states and from research institutions to compile the historical records.

Response: This line is improved.

Comment: Page 2, line 11 – The most recent version of the NOAA record is now called NOAAGlobalTemp with the following reference: …..

Response: NOAA record name is changed and two references are cited.

Comment: Page 2, line 14 – Ishii et al., 2005 describes only the marine portion of the JMA temperature record.

Response: Ishii et al.,2005 is replaced with Ishihara (2006).

Comment: Page 2, line 15 – The latest version of HadCRUT, HadCRUT5, has the following reference. Note the date for the final published version as 2021, not 2020:

Response: HadCRUT5 reference is added.

Comment: Page 2, line 15 – The 2013 paper for Berkeley Earth only described the land data. The recent paper describing the merged land-ocean can be cited as:

Response: Rohde et al 2013 is replaced with Rohde and Hausfather (2020)

Comment: Page 2, line 19 – The GISS data set has long only used satellite nightlight data for bias adjustment of urban areas. It does not use satellite derived temperature information. The current version of MLOST does not use satellite data. The statements here likely refer to the

ERSST3b sea surface temperature data set, which used satellite data and was once the marine data source for these data sets. The current version, ERSST5, does not use satellite data.

 Response: This line is modified accordingly.

Comment: Page 2, line 26 – HadCRUT4 is not interpolated but the recently publish HadCRUT5 is. The JMA data set's oceans are interpolated.

Response: This line is modified.

Comment: Page 2, line 30 – MLOST is not based on linear interpolation. It's a combination of "low frequency" spatial running average and a "high frequency" reduced space analysis using a method called Empirical Orthogonal Teleconnections.

Response: Thanks. This line is modified.

Comment: Page 2 line 31 – GISS uses linear distance weighting, not inverse linear distance weighting. No inverse involved (see Equation 1 of Lenssen et al., 2019 for the equation, or section 2 of Hansen et al., 2010 for a description). The linear distance weighting is correctly stated in the following sentence on line 32.

Response: The word "inverse" is removed. By linear inverse distance weighting we mean that the weight of each sample point decreases linearly from unity to zero. It does not explicitly include inverse. This interpolation scheme computes estimates by weighting the sample points closer to the prediction location greater than those farther away without considering the degree of autocorrelation for those distances.

Comment: Page 2, line 34 – A reference is needed here. Does this use of kriging refer to the JMA COBE SST data set's use of optimum interpolation?

Response: Yes. The reference and word ocean are added.

Comment: Page 3, line 1 – Cowtan and Robert (2014) should be Cowtan and Way (2014) (i.e. the author's name is Robert Way). Repeated again in other references to this paper (e.g. on line 3).

Response: Thanks. This reference appears in two lines that are corrected.

Comment: Page 3, line 4 – I would argue that these methods do not ignore variations at multiple length scales. For kriging/Gaussian process regression, the ability to represent multiple length scales is dependent on the covariance function used (which can be extremely flexible if constructed to be). The reconstruction method in NOAAGlobalTemp also represents multiple length scales in its own way though a reduced space decomposition. I assume that this comment on multiple length scales is alluding to MLRK, which explicitly represents multiple length scales as a sum of covariance functions. The distinction here is perhaps in MRLK flexibly to fit covariance structures with multiple scales without necessarily defining those structures in advance?

Response: This line is removed from this paragraph as the MRLK feature of capturing variations at multiple scales is discussed in the next paragraph (line-8).

Comment: Page 3, line 5 – The new HadCRUT5 data set include a conditional simulation step to encode analysis uncertainties into an ensemble. Other data sets provided uncertainty estimates in their interpolation by other means but not through simulation.

Response: This line is removed.

Comment: Paragraph at page 3, line 15 - It is great to see the hyperparameter uncertainties and conditional simulation included. It does not appear that all components of the HadCRUT4 uncertainty model have been used though. In particular, those associated with biased observations for individual ships encoded into the HadCRUT4 error covariance matrices and per-grid-cell uncertainty estimates, not included in the ensemble, are not used. Instead the model seems to assume i.i.d. errors for each observed grid cell, with a stationary measurement error variance across all locations, estimated each month.

Response: Page-3 line - 17 is removed and a line is added at the end of this paragraph.

Comment: Page 3, line 27 – Should this comment on sparsity in covariance matrices refer instead to sparsity in inverse covariance matrices?

Response: Yes. Covariance is replaced with precision in this line.

Comment: Page 4, line 19 – I know that I can refer to Nycha et al. (2015) to understand the function of the lambda parameter. The average reader will not know this. A reference to Nycha et al. (2015) would be appropriate here. Is "aw" one term or two? I don't understand what this parameter does and I can't find it in Nycha et al.

Response:  Smoothing parameter lambda is explained in this paragraph. "aw" is also explained in this paragraph and is replaced with a.wght.

Comment: Page 4, line 21 – Typo? "The smoothness parameter lambda influence throughout the calculation". What does is it influence?

Response: This section of the line is removed.

Comment: Page 5, line 12 – Are these semivariances defined at the observation locations. Is there any binning etc. to compensate for biases sampling of e.g. short ranges when computing the empirical semivariogram?

Response: These details are added next to line-12.

 Comment: Page 5, line 4 – d and rho are not defined in this paper.

Response: These are defined now.

Comment: Page 5, line 10 – version 4.5.0.0 but it's appropriate for comparisons with Ilyas et al. (2017) if it is the same version as used there.

Response: This paper and Ilyas et al. (2017) are based on the same version 4.5.0.0.

Comment: Page 7, line 6 – This Section 3.2 is essentially a recap of Morice et al (2012), with a heavy focus on the land data. Only the large-scale bias terms are discussed here and not the measurement and grid sampling uncertainty components. These are particularly important for

marine regions as ship/buoy movement leads to spatially correlated error, which should be mentioned here. HadCRUT4 does not include these in the ensemble, but instead as additional spatial error covariance matrices. It seems that these have not been used in this paper.

Response: Only ensemble data of HadCRUT4 is used in this paper. Gridded error variances of HadCRUT4 are not used. This is mentioned at the end of this section.

Comment: Page 7, line 3 – The sentence should not begin with "So".

Response: This line is modified.

Comment: Page 7, line 24 – It could be noted that the error model here represents the effects of potential residual biases when using station records that have been screened for urbanisation.

Response: This is added.

Comment: Page 7, line 28 - Sampling distributions for the HadCRUT4 ensemble are described in Morice et al. (2012). It would be sufficient here to refer to that study for the ensemble sampling methods rather than repeating it here and elsewhere in Section 3.2. The output of Morice et al. (2012) is used in this paper rather than reimplementation/modification of their methods so these methodological details are not core to this study.

Response: Line-28 and line-29 are removed. Additionally, line-15 and line-20 are also removed.

Comment: Page 8, line 4 – Again, sampling distributions used in the construction of the input datasets used could be replaced with a reference to Morice et al. (2012) as they are not critical to the new work undertaken in this study.

Response: This line is excluded.

Comment: Page 8, line 20 – Estimation of hyperparameters for each individual choice is an important design choice. There is no discussion of variation of the parameter estimates seasonally or in time later in the paper. It would be interesting to see this.

Response: Two more figures are added in the Appendix and discussed in Section 4.1.

Comment: Page 8, line 24 – this is 10 hyperparameter sample draws for each month, yes?

Response: That's true. Word hyperparameter is added.

Comment:

Page 28, line 28 – do these marginal variances have interpretable units? Are these °C^2? It's interesting if the marginal variance has little affect on uncertainty as it controls the variance of the process in interpolated regions and the relative importance of each spatial scale. Or is the uncertainty in process variances somehow pushed into lambda in the model's parameterisation? The next sentence says that this parameter is computed from a single field. Is there no seasonal variability in marginal spatial variance? Perhaps some comment on how the parameter should be interpreted would be helpful to explain why February 1988 is representative of the whole data set.

Response: Alpha is unitless as it is a weight. The alpha is a vector that is the relative weight given to each layer of the multiresolution model. And each layer is normalized to have a marginal variance of 1. The overall "rho" is then the complete marginal variance of the process.
That way the units in rho are squared temperature units.

We estimated the marginal spatial variance for the field that has the maximum information. This slightly varies across time. The parameters found (denoted as alpha in the LatticeKrig model) for February 1988 are the relative variances between different resolution levels. The decay of these for finer resolution fix the smoothness of the field and because these are difficult to estimate they were not emphasized in the analysis and the correlation range parameter, a.wght, was given more attention. February 1988 has the most complete data field and we assume that the smoothness reflected in the alpha parameters are consistent across different times. Beyond that we agree with the reviewers tacit point that this may not be completely representative of all months.

Comment: Page 9, line 2 – Use of the field with minimum coverage seems a strange choice rather than using a well sampled period. Is this because of computational cost limitations?

Response: Not really. We intend to show the coverage error estimates of the least sampled spatial field. Presumably, this field was expected to demonstrate the advantage of using a fancier spatial model. Therefore, the posteriors are being shown for the corresponding spatial field to develop a link. This is explained towards the end of this section .

Comment: Figure 2 - It would be interesting to see the likelihood-based estimate here too. This would help to understand what's happening in Figure 3 in comparisons between ABC and likelihood-based fields. This figure would benefit from some accompanying discussion of the effects of the parameters on the fitted fields and how the parameter estimates differ from the likelihood-based parameters.

For example, Nychka et al (2015) indicates that lambda =noise variance / process variance. High values of lambda would give a process with low variance and large measurement noise. Would this result in e.g. lower variance fields than a likelihood based estimate of a lower lambda?

Response: Second paragraph in Section 4.2 is added for this.

Comment: The aw parameter is interesting here. There's a lot of weight right at the edge of the prior distribution. Is the distribution being truncated by the choice of prior?

Response: Not really. The choice of prior was guided by Nychka et al. (2015). Details are added to the first paragraph of section 4.1.

Comment: Page 9, line 11 – Some discussion of how parameters compare between ABC and likelihood methods would again be useful here. What is it about the sampled parameter values that leads to the differences?

Response: These details are added to the second paragraph of section 4.2.

Comment: Page 10, line 2 – The lower uncertainty in unobserved grid locations could explain why This could be the reason that global average statistics do not appear to be much affected.

It seems like the ABC ensemble is leaning towards a model with greater observational noise variance and lesser process variance. This would explain the smoother fields in the figure 2 (a)-(b) comparison. Is this correct?

Response: This can be a reason.

Comment: Figure 3 – Are these fields the ensemble means/medians? This is not stated. Are the results for this month representative of other months in terms of parameter estimates and uncertainty estimates? It would be useful to see how they compare for better sampled periods or other times of year. The plot needs units (°C?).

Response: The field is ensemble median. It is mentioned on page-9 line-2. Units are added as well. Spatial estimate and uncertainties for two more spatial fields are presented in Figure B2 to Figure B4 in the appendix.

Comment: Figure 4 – How is uncertainty defined here? Is this the full ensemble spread? What is the statistic being shown? What are the units?

Response: Figure 4a represents the gridded standard error in °C associated with the gridded predictions made in Figure 3a. The predictions are for the median ensemble member. These details are mentioned in the caption of the figure.

Comment: Page 12, line 5 – I think that this says that samples are drawn from the conditional normal distribution, with each HadCRUT4 ensemble member having 10 hyperparameter samples, and each of those having 100 samples from the conditional normal. However, the wording of "namely the variogram based ABC posteriors of autoregressive weights and smoothing parameter" does not include the conditional normal sampling.

Response: That's true. Additionally, drawing posteriors does not include the conditional normal sampling.

Comment: Figure 5 – Is there a grey line plotted for the ensemble median of the old ensemble? If so then I can't see it. It would be helpful to include to show any differences (or lack thereof) in the mean/median with the hyperparameter sampling.

Figure needs units (°C?).

Response: Old ensemble median time series is superimposed. Unit is added as well.

Comment: Page 14, line 5 – It's not clear that the uncertainties are always larger for ABC. They seem comparable or slightly skewed relative to Ilyas et al 2017. Differences often appear to be around the scale of the apparent rounding resolution in the plot. It looks like the uncertainty range is often narrower for the new ensemble.

Is it guaranteed that the uncertainty estimates would be wider using ABC? Could lower uncertainties be possible if the hyperparameter distribution samples a region of the parameter space that leads to smaller process variance, and hence smaller coverage uncertainty estimates?

Response: This line is removed and details are added.

Comment: Page 14, paragraphs at lines 13 and 24 – This discussion of Latin hypercube sampling is a strange and abrupt way to end the paper. It would better be placed earlier in the

method/results. This sampling is good to see though as a 100,000-member gridded dataset would be rather unwieldy to use.

Response: These paragraphs are moved earlier

Comment: Page 15, line 1 - No conclusions section? See main point 6.

Response: Conclusion section is added.

Comment: Figure 6 – The axes need labels/units. It could be useful to see the resulting sampling for other locations too.

Response: Axes labels/units are added. Figures for other months are added.

**Referee-2:**

**Assessment:** In the end, it seems to me that the authors come out with an incremental improvement on their own previous construction and it doesn't seem to be the "game changer" that the authors were possibly hoping for when they set out along this path. For example, the uncertainty bounds in Figure 5 are only very slightly different from those in their own previous construction, and the extended example on pages 9-11 seems to have been specially constructed as a worst-case scenario (as is explained in the paper, May 1861 was the month with the least spatial coverage of the entire dataset, therefore, presumably, the one for which the advantages of this kind of approach should be most clearly seen). Nevertheless, the work seems to be been competently carried out and it is always useful to have another dataset for comparison - I fully support publication.

Comment: My main concern about the paper is that the manuscript itself seems to have been put together in some haste. Prior to reading this, I was familiar with the LatticeKrig approach but not with the intimate details. Some of those details are important, e.g. the definitions of the lambda and aw parameters (page 4). I believe, throughout the paper, there is a need to give more explicit detail about the method. Given the 62 months of computer time, it seems unlikely that anyone else would want to exactly reconstruct this dataset, but nevertheless, I still feel that this should be a requirement of publication, that the method should be described in sufficient detail that anyone who wants to reconstruct the result has all the information required to do so.

In my initial correspondence with the editor, I queried why the authors had not produced a "supplemental materials" document - as I understand, such a document would be supported by the journal and I might suggest that the authors take advantage of this when revising the paper. Specifically, the kind of detail that is probably not relevant to the casual reader of the paper, but would be needed by anyone actually intending to try to reproduce the results, could very well go into an online appendix.

Response: Apologise for not properly defining the model parameter definitions. These details are added now.

Comment: One further "general" comment - there is a passing reference to HadCRUT5 (Morice et al. 2020), which is the latest version of the U.K. Met. Office model. I wonder if maybe the authors should say a little more about this. I presume the production of HadCRUT5 overlapped the present effort but would the authors like to comment on how HadCRUT5 improves on HadCRUT4 and specifically how it compares with the present work?

Response: More details of HadCRUT5 are added in section-3.

**Specific comments:**

Comments: p. 1 line 16 - "phenomena" (plural of phenomenon)

Response: Done.

p. 2 l. 15 Here the author Rohde is misspelled Rhode. This error also occurs in p. 2 l. 20, p. 3 l. 1 and p. 18 l. 24

Response: Done.

p. 2 l. 34 Reference to JMA (?) where the question mark is a standard latex warning for a missing reference. Were the authors referring to the paper Ishii et al, mentioned on p. 2 l.14?

Response: Apologies. This is corrected.

Comments: p. 4, l. 19. Here the authors mention two parameters from Nychka et al 2015, called lambda and aw, but they never define these two parameters. The implication is that one can look up Nychka 2015 to find these definitions but I tried doing that and I think we need more assistance.

Nychka (2015) defines a parameter lambda=sigma^2/rho but they don't call it a smoothing parameter - that was my first confusion. I do note that the LatticeKrig R manual also defines lambda and does call it a smoothing parameter - the most recent version of this that I downloaded for preparing this review was version 8.4 dated November 2019 (the authors of this paper refer to version 6.4 as the one they used for the bulk of their computations). If the present authors want to call it "the smoothing parameter" without further explanation, they need to be precise about where this is defined, and the answer appears to be the LatticeKrig manual (which I'll subsequently refer to as LK), not Nychka (2015) (henceforth N15).

Response: The LatticeKrig manual reference is added.

Comments: Both N15 and LK say that lambda and rho are computed my maximum likelihood and I understand that one of the objectives of the present paper is to extend that by using the ABC approach to approximate lambda, but what happened to rho? This isn't explained here, but later (p. 6 l. 4) they say, "both d and rho are still estimated using the maximum likelihood approach".

In fact d and rho are respectively an overall mean and a variance (scaling) parameter of a multivariate normal distribution and it is well known and trivial to implement that these parameters can be integrated out analytically, so as to focus attention on the spatial correlation parameters - in my first reading of the paper I assumed this was what they had done. But the comment on p. 6 l. 4 makes me wonder about this point. In summary, we need clarification of what the authors actually did. (If it really was a Bayesian approach, we also need to discuss the prior distribution, since rho in particular may require a proper prior.)

Response: Apologies for this confusion. Only ABC posteriors of lambda and autoregressive weights were being determined. Other parameters (e.g. d and rho) were estimated by maximum likelihood. This is explained in the paper, now.

Comments: Now let me turn to the other parameter, aw, referred to as "autoregressive weights". This is based on the fact that at each level of the multiresolution process that defines the spatial model, the coefficients of the radial basis functions have the structure of a lattice process that is assumed to be of conditional autoregressive (CAR) form. However, here is no fixed structure for this and no single parameter called the autoregressive weight (or weights - it's not clear whether the authors actually meant to use a plural form here). p.584 of N15 refers to a weight matrix B where the off-diagonal entries are -1 and the diagonal entries are of form $B_{j,j}=4+kappa^2$. In this case the lower bound 4 arises because the sum of entries in B is required to be positive. So is kappa the autoregressive weight? LK actually use a different notation, where they define a variable a.wght (this is the nearest I can find to any variable actually called "aw") and they comment (p. 23), "in the simplest case a.wght is the central value, and should be greater than 4". So is a.wght the same as $4+kappa^2$ in the N15 notation? If so, which parameterization do the present authors actually use? Later (p. 8 l. 21) the authors give aw a prior density that is uniform on the interval (1,4), but now I'm really confused about how that particular range was determined …

Another potential wrinkle is that N15 p. 585 explicitly mentions the possibility that the autoregressive structure may be different at different levels of the multiresolution process, but I'm reading between the lines that they didn't consider that extension in this paper.

I don't actually think any of these questions are complicated. I understand very well that there are certain model choices that you just have to make. The authors simply need to be explicit about what these choices were and how exactly the various parameters are defined.

Response: The reviewer is well justified in asking for clarification about the exact form for the spatial autoregression (SAR) parameter that controls the spatial dependence among lattice points. Although there is the freedom to choose a different parameter for each level of multiresolution level in this work it is fixed at a single value. This  choice is consistent with more traditional covariance models such as the Matern family in spatial statistics. As the reviewer correctly pinpoints, the a.wght parameter built into the LatticeKrig code for rectangular lattice points is the autoregressive parameter and is the same as $4+kappa^2$ in 2 dimensions and in this form the 4 nearest neighbors get a weight of -1 in the SAR formulation. Moreover, kappa can be loosely interpreted as a correlation range parameter for the spatial field. The particular application to a SAR model on the sphere and the icosohedral based lattice, however,  led us to a slightly different parameterization. This is due to the fact that the lattice points can have different numbers of nearest neighbors. Twelve points will have 5 neighbors and the remaining points will have 6.  Moreover,  these points will not be exactly equally spaced. Given these irregularities we found it easier to give a weight of $1 + exp(2*omega)$ to the center  lattice point and constrain the sum of  the nearest neighbor weights to sum to negative one.  Thus, in this spherical version we take a.wght= $= 1 + (exp(omega)^2 = 1 + exp(2*omega)$ and parametrize this value through omega. exp(omega) can be interpreted as an approximate correlation range for the SAR and omega itself is useful in representing the log of this scale parameter.   In particular priors put on a.wght are done through the assumption that omega follows a uniform distribution over the interval [ -4.5, .55].  This choice covers a useful span of spatial correlations when omega is translated back into the a.wght parameter and subsequently into the dependence of the field at the lattice points.

Comments: p. 5. The flow diagram illustrating the ABC approach is clear and should be easy for the reader to follow, but again, some specific details are missing. How do they determine the number H of variogram sampling points and the specific vector of distances represented by h? I'm assuming that when you write gamma(h) you don't literally mean h as the distance (h is

an index going from 1 to H) but gamma(d_h) for some distance d_h, but then the same question, what values did you actually use and how were they chosen?

Response: All the standard rules are observed while computing the semivariogram. Semivariogram is computed up to half of the maximum distance between the points over the whole spatial domain. The distances are Great circle distances in kilometers. Number of bins is 12 that is somehow standard. "h" is the center of each of 12 lag classes that are defined over half of the maximum distance. These details are added to the paper.

Comments: p. 6 equations (5) and (6). I don't think we need you to define every symbol here but please give an exact source for these equations, which I assume are somewhere in N15?

Response: These equations are slightly different from those in N15.

Comments: p. 6 lines 16,17 - reference to a new dataset HadCRUT5 (which I wasn't aware of myself until reading this paper). I presume HadCRUT5 came out while this paper was being developed. I think this sentence should be moved to the discussion section and the authors should discuss how the two approaches compare and contrast with one another - are there any features in which HadCRUT5 improves on the present approach?

Response: These details are added in section-3.

Comments: p. 7, l. 22 - "since the last few decades" - slightly awkward English construction here, maybe "during the last few decades" would be better. I am aware that the word "since" would be used in several other languages, for example "depuis" in French.

Response: Thanks. Replaced.

Comment: p. 7, l. 28. Unclear why you set any negative value to 0.0. While I'm well aware that we all talk about global warming and not global cooling, I don't think the possibility of cooling is excluded by basic atmospheric physics - if a stochastic model occasionally produces a negative value, why not include it in the analysis? From a political point of view, the authors should take care to avoid any implication that their approach was predetermined to result in a warming outcome.

Response: These details are from Morice et al. (2012). We have removed these sampling details from this paper in light of the comments from other reviewer.

Comment: p. 8 l. 28. This review is already getting rather lengthy and by this point I was definitely suffering from reviewer fatigue, but if I'm not mistaken, this is the first time in the paper there is any parameter called alpha. Please, either define it, or give an explicit prior reference where it is defined.

Response: Reference is cited.

Comment: pp. 9-11. The authors are quite explicit that May 1861 was chosen for this illustration because it was the month with the poorest spatial coverage, and therefore presumably the one that best illustrates the advantages of using a more refined spatial approach, but I think it would be helpful to have at least some comparisons with other months. Are these kinds of plots typical of what we would expect if we just chose a month at random?

Response: More plots are added in the appendix to clarify this.

Comment:p. 10 lines 3-4: so part of the reason for the difference is that the LKrig function was improved between the two versions of LatticeKrig that were used for the 2017 paper and this one? Could you expand on that a bit - was that a major factor?

Response: More details are added in section 4.2 after the first paragraph.

Comment: p. 12 l. 24 and Figure 5b: what exactly is the "median time series"? I'm inferring that each section of the time series was centered about its median value but what time scale was used for calculating the medians?

Response: For calculating the medians annual time series were used.

Comment:p. 14, this is an additional feature that is only introduced later in the paper and somewhat complicated to evaluate. It seems that the authors do not intend to publish their full 100,000-member ensemble but only a subset selected by a conditional latin hypercube sampling (CLHS) approach? I'm sure there are good reasons for doing that but at least from the appearance of Fig. 6, there appear to be some nontrivial differences between the two approaches, or am I misinterpreting this figure? Once again, the fact that they have shown this figure only for May 1861 may in some sense be a worst case scenario, but it would be helpful to clarify that point.

Response: We would be happy to publish the full ensemble. We are looking at the options. Also, in the appendix figures for other months are added.

---

## Author Response (AR2)

**Response to Reviews**

Major comments

1. The additional discussion in the conclusions section helps to put this work into context of the current state of understanding in the field. The paper's conclusions are at times perhaps overly negative about the value of the study. That the inclusion of the parameter ensemble does not have a substantial impact on uncertainty in global average temperature estimates is a useful result. The paper's results indicate that parameter uncertainty estimates may be important at regional scales. The true impact of this will not be clear until the data set is used in other studies.

Response: This section is slightly revised in light of these comments.

2. There appears to be no reference to Figures A1 and A2 in the text. Figures C1 and C2 appear to be referenced in the incorrect location (where it seems that A1 and A2 would be relevant).

Some additional comment on what we see in the figures in Appendix A and Appendix B would also be beneficial as they appear to show that the parameter estimates differ for differing months. This appears to have an impact on uncertainty estimates in less well-measured regions.

Response: These issues are addressed now.

3. There are several typos and grammatical errors to be corrected (see detailed comments).

Response: These are corrected now.

Detailed comments (line numbers corresponds to the version with tracked changes):

➢ Page 1 Line14 – I suggest removal of the word natural from this sentence. The cited references assess observed variability with no such assessment on whether this variability is natural or non-natural (i.e. anthropogenically forced).

   Response: Many thanks, word natural is removed.

➢ Page 1 line 15 –. A slightly odd choices of references with limited relevance to this study. The use of model here is rather broad, covering two assessments of NWP model output and an application of an integrated assessment model. Milton and Earnshaw (2007) compare NWP temperature output to individual observation sites and not global gridded data sets (although they do use spatially gridded data for precipitations). Edwards et al (2011) assess NWP temperatures against observations from only a single observing site (no spatial estimation or fields used). Glanemann et al. (2020) does use global temperature data from NASA GISTEMP.

   Response: Milton and Earnshaw (2007) and Edwards et al (2011) citations are removed from this line.

➢ Page 1, line 19 – To say this data is "mostly derived" from these sources may not be accurate. While much modern data is indeed shared under WMO and GCOS umbrellas, the historical station data sets depend greatly on data compilation by research and data rescue efforts. This includes work by groups at NOAA, the Climatic Research Unit, the International Surface Temperature Initiative, and others such as the ACRE initiative.

   Response: This line is modified.

➢ Page 2, line 29 – typo: does not.

Response: Corrected.

➢ Page 3 – line 10 – Could mention HadCRUT5 here, as this supersedes HadCRUT4 and includes spatial interpolation through kriging/gaussian process regression.

Response: Lines 1-4 on page-7 are shifted here. (previous track changes document)

➢ Page 3, line 26 – It has yet been mentioned that the paper uses HadCRUT4 so this line appears abruptly. Should this sentence appear later in the paper, in the first paragraph of Section 3 or at the end of section 3?

Response: This is shifted at the end of section-3.

➢ Page 4, line 27 – This additional description of the functions of the autoregressive weight and lambda parameters, with reference to Nychka et al. (2019) is appreciated.

Response: Many thanks.

➢ Page 5, line 2 – There are a few grammatical errors/typos in this added text: "*The* empirical semivariogram", "the average distance *between* each bin", "*The* semivariance", "The semivariogram".
➢ Page 7 line 20 – typos: "Therefore, it was was converted to an ensemble data *set* by Morice et al. (2012) using *the* Brohan et al. (2006) uncertainty model."

Response: Apologies, these are corrected now.

➢ Page 8, line 25-28 – this is now clear that only the ensemble is used in this study and not any additional uncertainty information provided in HadCRUT4.

Response: No action needed.

➢ Page 9, line 13 – should omega be a symbol here?

Response: We used the same notation as it is used in the manual of the LatticeKrig package so that the readers could relate easily. However, it is converted to maths writing now.

➢ Page 9, line 25- the additional detail on why 1988 was chosen is useful. The additional technical details in this section improve the reproducibility of the study. This will also be of aid to readers that are familiar with the LatticeKrig method.

Response: No action needed.

➢ Page 9, line 32 – sentence construction: please change "Other two" to "Two other".

Response: Thanks, changed.

➢ Page 10, line 1 – It is helpful to see the modelling assumptions now clearly stated here. The additional figures and discussion in section 4.2 are welcome additions that aid interpretation of the differences between the new results and those of Ilyas et al (2017).
➢ Page 14, line 15 – thank you for adding this discussion of the effects of lambda for different sampled values.

Response: No action needed.

> Figure 5 – It is rather hard to see the lines in this figure. Please consider increasing the contrast between the lines and shaded regions.

Response: Contrast is increased.

> Page 17, line 1 – This should mention the reason that the data set it is rather unwieldy – the large ensemble.

Response: Mentioned.

> Page 17 lines 11 – I recommend adding the word average here to be clear that this statement refers to global average temperature estimates. The case for regional temperature is not clear as it has not been investigated in detail.

Response: The word 'average' is added.

> Page 17, line 14 – Inconsistent use of capitals: HADCRUT4/HadCRUT4 (HadCRUT4 used elsewhere).

Response: This is corrected here.

> Page 17, line 15 onwards – That the inclusion of the parameter ensemble does not have a substantial impact on uncertainty in global average temperature estimates is a useful result. It indicates that the use of point value estimates in existing assessments does not have a substantial impact on total uncertainty estimates. The comment on use in studies of climate variability at line 15 is apt. While not investigated in detail in this paper, it is quite possible that the interpolation parameter uncertainty may be more important for regional climate studies than for the global average.

Response: No action needed.

> Page 17 line 17 – This statement here may be over critical of the value of the work. Spatial model parameter uncertainty estimates may be important for regional assessments, which is not ruled out by this work.

Response: This line is removed and two lines are added.

> Appendices – These figures may be better suited to a supporting information section rather than appendices, dependent on the Journal's style guidance.

Response: Noted.

> Figures A1 and A2 - Figures A1 and A2 do not appear to be cited in the text. Should the references to Figures C1 and C2 at page 9 line 32 refer to these two figures? These two figures with Figure 2 suggest that the model parameters do change for different fields/times. The corresponding uncertainty fields in Figures 3, B1 and B2 suggest an impact of uncertainty in spatial fields, particularly in unobserved regions. This does not appear to be commented on in section 4.2.

Response: These details are now added in section 4.2. Page-9, line-32, reference of the figures is corrected as well.

> Figures C1 and C2 – These appear to be referenced in the incorrect location in the text (see previous comment). They should be referred to in Section 6, around page 16 line 4.

Response: Thanks, these are referred in this section now.